# Intrahippocampal Inoculation of Aβ_1–42_ Peptide in Rat as a Model of Alzheimer’s Disease Identified MicroRNA-146a-5p as Blood Marker with Anti-Inflammatory Function in Astrocyte Cells

**DOI:** 10.3390/cells12050694

**Published:** 2023-02-22

**Authors:** Ruth Aquino, Vidian de Concini, Marc Dhenain, Suzanne Lam, David Gosset, Laura Baquedano, Manuel G. Forero, Arnaud Menuet, Patrick Baril, Chantal Pichon

**Affiliations:** 1Centre de Biophysique Moléculaire, CNRS UPR 4301, Rue Charles Sadron CS 80054, CEDEX 02, 45071 Orléans, France; 2Faculty of Science and Philosophy, Universidad Peruana Cayetano Heredia, Lima 4314, Peru; 3Faculty of Science and Techniques, University of Orléans, 45067 Orléans, France; 4Experimental and Molecular Immunology and Neurogenetic, UMR7355 CNRS, 45071 Orléans, France; 5CEA, CNRS, Laboratoire des Maladies Neurodégénératives, Université Paris-Saclay, 18 Route du Panorama, 92265 Fontenay-aux-Roses, France; 6Professional School of Systems Engineering, Faculty of Engineering, Architecture and Urban Planning, Universidad Señor de Sipán, Chiclayo 14000, Peru; 7Institut Universitaire de France, 1 rue Descartes, 75231 Paris, France

**Keywords:** microRNAs, miRNA-146a-5p, Alzheimer’s disease, Aβ_1–42_ peptide, biomarkers, diagnosis

## Abstract

Circulating microRNAs (miRNAs) have aroused a lot of interest as reliable blood diagnostic biomarkers of Alzheimer’s disease (AD). Here, we investigated the panel of expressed blood miRNAs in response to aggregated Aβ_1–42_ peptides infused in the hippocampus of adult rats to mimic events of the early onset of non-familial AD disorder. Aβ_1–42_ peptides in the hippocampus led to cognitive impairments associated with an astrogliosis and downregulation of circulating miRNA-146a-5p, -29a-3p, -29c-3p, -125b-5p, and-191-5p. We established the kinetics of expression of selected miRNAs and found differences with those detected in the APP_swe_/PS1_dE9_ transgenic mouse model. Of note, miRNA-146a-5p was exclusively dysregulated in the Aβ-induced AD model. The treatment of primary astrocytes with Aβ_1–42_ peptides led to miRNA-146a-5p upregulation though the activation of the NF-κB signaling pathway, which in turn downregulated IRAK-1 but not TRAF-6 expression. As a consequence, no induction of IL-1β, IL-6, or TNF-α was detected. Astrocytes treated with a miRNA-146-5p inhibitor rescued IRAK-1 and changed TRAF-6 steady-state levels that correlated with the induction of IL-6, IL-1β, and CXCL1 production, indicating that miRNA-146a-5p operates anti-inflammatory functions through a NF-κB pathway negative feedback loop. Overall, we report a panel of circulating miRNAs that correlated with Aβ_1–42_ peptides’ presence in the hippocampus and provide mechanistic insights into miRNA-146a-5p biological function in the development of the early stage of sporadic AD.

## 1. Introduction

Alzheimer’s disease (AD) is a complex and multifactorial pathology that affects millions of people around the world [1]. The appearance of amyloid plaques and the formation of neurofibrillary tangles (NFTs) are two representative features of AD, responsible for the gradual deterioration of cognitive functions such as loss of memory, language, and thinking ability. Amyloid plaques are deposits of amyloid beta (Aβ) peptide that accumulate in the extracellular matrix between nerve cells [2]. The Aβ peptides arise from the cleavage of the amyloid precursor protein (APP). Among the different Aβ species generated, Aβ_1–42_ peptides are the most hydrophobic and fibrillogenic and are the main species deposited in the brain [3]. Neurofibrillary tangles are intraneuronal acumulation of hyperphosphorylated tau protein (41) [4].

One of the main issues with AD is the long preclinical stage. This pathology begins 10–20 years before significant neuronal death and cognitive symptoms and behavior defects appear [5]. Although much is known about the disease, the early molecular events that trigger the pathogenesis of AD are not yet fully understood. It has been postulated that Aβ peptides, especially Aβ_1–42_ peptides, which are more prone to aggregation, initiate a cascade of pathological events that lead to aberrant phosphorylation of Tau, neuronal loss, and eventual dementia [6,7,8]. The aggregation of Aβ peptides can generate oligomers and fibrils that can both be neurotoxic. Today, there is no therapy able to cure or prevent AD [9]. There is a real need to predict the development of symptomatic AD for both mild cognitive impairment (MCI) and dementia in asymptomatic individuals [10]. The current AD diagnosis uses cumbersome and expensive methods such as structural magnetic resonance imaging (MRI) and molecular neuroimaging with positron emission tomography (PET) [9]. Therefore, the search for biomarkers for early diagnosis is still essential and is currently an active field of research.

MiRNAs are a subclass of small noncoding RNAs that play important roles in the regulation of post-transcriptional gene expression by binding to complementary sequences of target messenger RNAs (mRNAs) and inducing translation repression and/or mRNA degradation [11,12]. Under different pathological conditions, there is a dysregulation of miRNAs present in most body tissues and fluids, including brain tissues, cerebrospinal fluid (CSF), and serum [12]. MiRNAs present in biofluids are called circulating miRNAs [13]. They are produced and secreted from cells present in tissues and organs, and so reflect the pattern of expression of the tissues of origin. Circulating miRNAs are now considered as disease biomarkers [14]. They are stable in different biological fluids, relatively easy to detect, and have an expression pattern that reflects the disease stages of pathologies such as AD [15,16,17]. All of these characteristics position miRNAs as potential AD biomarkers.

Several animal models have been created for AD research, with transgenic animal models being the most popular [18]. As with any animal model, they must mimic all the cognitive, behavioral, and neuropathological characteristics of the disease to recapitulate the disease phenotype with high fidelity. Unfortunately, most of the AD models currently used are called partial models, as they mimic only some components of AD. Amyloid-producing transgenic animal models of AD have been created based on the genetic origins of familial AD or early-onset AD (EOAD), detected in patients under 60 years of age, which corresponds to only 5% of AD patients. These models have contributed significantly to better understanding the molecular mechanisms involved in the pathology, but they do not represent the majority of AD cases, called late-onset AD (LOAD), which represents 95% of cases that develop in patients over 60 years of age [19,20,21]. In addition, the different genetic backgrounds of these models constitute a real issue [22]. Due to these drawbacks, non-transgenic animal models have been developed to study the most common LOAD form of AD [21]. These models represent one or more distinctive features such as AD-like senile plaques, NFT, oxidative stress, and cognitive impairment [22]. Amongst them, one model consists in injecting an Aβ synthetic peptide into the brains of rodents. Previous studies have shown that this procedure causes learning and memory deficits in treated animals due to the formation of amyloid plaques and the disruption of long-term potentiation and behavior [22,23,24]. As for the transgenic model, this model is not perfect either, but it provides some insights into the early impact of Aβ peptides on AD pathogenesis.

Different studies have investigated the expression of circulating miRNAs in the blood of AD patients and/or in transgenic animal models of AD. Results indicated that the most representative dysregulated miRNAs are miRNA-9a-5p, -146a-5p, -29a-3p, -29c-3p-125b-5p, -181c-5p, -191-5p, -106b-5p, and -135a-5p [25,26,27,28]. Those miRNAs have been associated with different stages of the progression of AD (for a review, see [29]). To the best of our knowledge, no studies have searched for the expression of a panel of several circulating miRNAs in rodent models of AD generated by intrahippocampal infusion of fibrillary forms of the Aβ_1–42_ peptide. Yet, it might be relevant to correlate them with a specific pattern of miRNA expression produced from cerebral adult tissues in response to the deposit of Aβ_1–42_ peptides.

This work performed with an AD model generated by intrahippocampal injection of fibrillary forms of Aβ_1–42_ peptide aims to correlate the expression of particular miRNAs with the presence of aggregated Aβ_1–42_ in the brain of an adult rat. A Morris Maze Water test was applied to assess the cognitive deficit of the rat model, while immunohistochemical analysis of GFAP expression was conducted to evaluate the presence of astrogliosis in brain tissues. Then, we quantified the amount of circulating miRNA selected from a short list of the most relevant miRNAs found in the AD literature. The same quantification was performed with the sera of amyloid-bearing APP_swe_/PS1_dE9_ transgenic mice for comparison between these two different animal models of AD. Last, we focused our attention on miRNA-146a-5p, as this miRNA was found exclusively differentially dysregulated in sera of an FAβ-infused rat model of AD and also because the kinetics of its expression in these animals were different from other established miRNA kinetics. We conducted functional in vitro studies on rat primary astrocytes treated with the Aβ_1–42_ peptide. We investigated the involvement of the NF-κB signaling pathway on the induction of miRNA-146a-5p expression and, as a consequence, the regulation of Irak-1, Irak-2, and Traf-6 as transcriptional targets of this miRNA and as functional effectors of the NF-κB signaling pathway. Finally, we evaluated the impact of this axis of regulation on the production of pro-inflammatory cytokines and chemokines such as *IL-1β*, *IL-6, TNF-α,* and *Cxcl1* genes in astrocytes in response to treatment with Aβ_1–42_ peptides.

## 2. Materials and Methods

### 2.1. Animals

Sprague–Dawley rats (aged 8–12 weeks) with a body weight of 200–380 g were obtained from the Bioterium of the Research and Development Laboratory located in the Cayetano Heredia Peruvian University. Animals were maintained under controlled laboratory temperature (25 ± 2 °C) and humidity (60%) conditions, with a controlled light cycle (12 h light/12 h dark). Water and food were available ad libitum throughout the experiment. Ethics Committee of the Cayetano Heredia Peruvian University (CIEA-102069) approved the animal handling and experimental procedures. The animal care staff monitored the behavior of rats daily to ensure that the animals were safe and healthy. APP_swe_/PS1_dE9_ transgenic mice and their littermate mice were bred and hosted in the animal facility of Commissariat à l’Energie Atomique (CEA, Centre de Fontenay-aux-Roses; European Institutions (Agreement #B92-032-02)). These mice express a chimeric mouse/human APP with the Swedish mutation and a human presenilin-1 lacking exon 9. All experimental procedures were conducted in accordance with the European Community Council Directive 2010/63/UE and approved by local ethics committees (CEtEACEA DSV IdF N°44, France) and the French Ministry of Education and Research (APAFIS#21333-2019062611099838v2).

### 2.2. Preparation of Amyloid-Β_1–42_ Peptide for In Vivo Use

Amyloid-β_1–42_ (Aβ_1–42_) peptides were from Sigma-Aldrich (Sigma-Aldrich, Saint-Quentin-Fallavier, France). The dry powder was solubilized in DMSO to generate a stock solution of Aβ_1–42_ peptides at concentration of 10 µg/µL (2.22 mM) in PBS. This stock solution was stored at −20 °C until use. For production of fibrillary forms of Aβ_1–42_ peptides (FAβ_1–42_), the working solutions were diluted in PBS at final concentrations of 0.5, 1, and 2.5 μg/μL and incubated for 5 days at 37 °C [30]. At the end of this incubation period, FAβ_1–42_ solutions were directly inoculated into the hippocampus of the rats as described below.

### 2.3. Model of Alzheimer Disease Generated by Intracranial Infusion of FAβ_1–42_

The general procedure to infuse the FAβ_1–42_ solutions in the brains of rats derived from Wu et al. [31] with some modifications. Briefly described, anesthetized rats were placed on a stereotaxic frame (KOPF^®^ 900, David Kopf Instruments, Tujunga, CA, USA) to infuse 3 μL of FAβ_1–42_ or PBS solutions into the two hemispheres of the hippocampus using a 10 μL syringe (Hamilton^®^ glass syringe 700 series RN, Hamilton, OH, USA) connected to a 26-G needle (Hamilton^®^). The procedure to inoculate FAβ_1–42_ and PBS solutions consisted of a gradual infusion of solutions over a 6 min period, followed by an additional 5 min period to ensure optimal dispersion of solutions into the ventricles. The Bregma was used as reference. The coordinates to infuse the solutions were as follows: 2.6 mm lateral, 3.0 mm back, 3.0 mm deep, corresponding to the CA1 regions of the hippocampus [32]. After careful removal of the syringe, rats were returned to cages and monitored every day until the time of experimentation.

### 2.4. Morris Water Maze Test

A Morris Water Maze test (MWM) was used to evaluate the spatial memory of animals. The general procedure was from Wenk et al. [33] with some modifications. A circular pool (126 cm in diameter, 75 cm in high) was built, filled with water at 21–22 °C, and loaded with a transparent plastic platform (10 cm in diameter) placed in a constant position. The pool was divided with imaginary lines to delineate four quadrants: northeast (NE), northwest (NW), southeast (SE), southwest (SW). Animals were placed in all quadrants, and their swimming trajectories to reach the platform were monitored using a webcam connected to a computer. Videos were processed with a home-made RatsTrack plug-in system, designed for these experiments to accurately record distance, time, and velocity of animals in the pool. The MWM was performed 14 days after the infusion of FAβ_1–42_ or PBS solutions and was monitored using an arbitrarily fixed time period of 90 s for each trial. A reference memory protocol consisting of familiarization, acquisition, and memory sessions was set up. The familiarization session procedure consisted in placing rats in one quadrant of the pool and to allow the rats to find the platform over 4 trials. The acquisition session was performed the day after the familiarization session and lasted over four days. The objective of this session was to evaluate the spatial learning of animals by placing rats in all 4 quadrants of the pool and evaluating their velocities to reach the platform. A total of 8 trials per rat were recorded. Finally, the memory session was performed on the sixth day. The objective of this trial is to record the reference memory of rats by monitoring the swimming trajectories of rats placed into the pool without any platform. One trial per rat was used in this specific session.

### 2.5. Body Fluid Collection and Sampling

#### 2.5.1. Blood Collection and Serum Separation

Blood samples were collected in BD Vacutainer^®^ tubes coated with silica as coagulation activator according to procedures described by Vigneron et al. [34]. Cardiac punctures were performed to collect enough blood per animal at day 7 (*n* = 8), 14 (*n* = 8) and 21 (*n* = 8) post-injection of FAβ_1–42_ or PBS solutions [35]. The samples were left at room temperature for 40 min for the formation of blood clot, and then centrifugated at 1900× *g* for 10 min at 4 °C to collect the serum. A second centrifugation at 16,000× *g* for 10 min at 4 °C was used to clarify the serum. Hemolyzed samples, inspected visually, were discarded from the study. The clarified sera were stored at −80 °C until used.

#### 2.5.2. Cerebrospinal Fluid Collection and Preparation

CSF was collected 14 days after the inoculation of FAβ_1–42_ solution from the cisterna magna of the rats according to procedure described by Pegg et al. [36]. Briefly, the procedure to extract CSF consisted of using an infusion system equipped with a 25-gauge butterfly needle to collect the CSF into the dura mater/Atlanto-occipital membrane. CSF samples (40 to 80 µL) were loaded in a 0.5 mL Eppendorf tube, incubated for 1 h at room temperature before centrifugation at 1000× *g* for 5 min at 4 °C to remove potential cell debris. A second centrifugation at 16,000× *g* for 10 min at 4 °C was used to clarify the samples. The clarified CSF samples were stored at −80 °C until use.

### 2.6. Analysis of Circulating miRNAs Expression in Animals

#### 2.6.1. Total RNA Extraction from Body Fluids

The NucleoSpin^®^ miRNA Plasma kit from Macherey-Nagel (Hoerdt, France) and the miRNeasy Serum/Plasma Kit (Qiagen, Hilden, Germany) were used to extract respectively total RNAs from sera and CSF samples. In the latter, 5 μg of a glycogen solution prepared at concentration of 0.1 µg/µL was added to each 50 µL of CSF fraction to optimize the yield of miRNA recovery [37]. Exogenous spike-in miRs (cel-miR 39-3p, -54-3p, and cel-miR 238-3p) were added to each sample to normalize the extraction procedure of miRNA according to standardized protocol described by Vigneron et al. [34]. Nucleotide’s sequence of spike-in miRs was taken from miRNA database and synthesized by Eurogentec (Eurogentec, Liège, Belgium) as SDS-PAGE purified oligonucleotides. These synthetic miRNAs were resuspended in nuclease-free water at a fixed concentration of 200 amol/μL. Then 2.5 µL per 100 µL of samples was added after the denaturation step of the extraction procedure, as recommended by the manufacturer’s instructions. Total RNA fractions were quantified using a nanodrop spectrophotometer (Nanodrop 2000, Thermo Scientific, Waltham, MA, USA). Samples with an RNA integrity number (RIN) greater than 8 were considered for the study.

#### 2.6.2. Reverse Transcription Reaction (RT)

Total RNAs were reverse transcribed by using the miScript II RT Kit (Qiagen, Hilden, Germany) according to routine procedure described in [38,39]. Fifty ng of total RNAs prepared at concentration of 10 ng/µL were polyadenylated by a poly (A) polymerase and then reverse transcribed to cDNA using oligo-dT primers following recommendations from the manufacturer (Qiagen). The generated cDNAs were then stored at −20 °C until use.

#### 2.6.3. Real Time Quantitative RT-PCR (qRT-PCR)

qRT-PCR was performed with the miScript SYBR^®^ Green PCR Kit (Qiagen, Hilden, Germany) according to routine procedures described by Simion et al. [38,39]. A volume of 2.5 µL of cDNA corresponding to 50 ng of cDNA was loaded into a final volume of 10 μL containing 5 μL of 2X QuantiTect SYBR Green PCR Master Mix, 1 μL of 10X miScript Universal Primer, 10X miScript Primer Assay, and 0.5 µL of RNase-free water. All miRNA-specific forward primers (miScript Primer Assays) were purchased from Qiagen and are listed in Appendix A. The quantification of PCR products was collected using the Light Cycler^®^ 480 (Roche Diagnostics Corporation, Indianapolis, IN, USA). The relative miRNA expression was calculated according to the Livak and Schmittgen method [40] and expressed as 2^−∆∆Ct^. The mean of Ct from the 3 spike-in miRNAs was used to normalize the data as described by Vigneron et al. and Faraldi et al. [34,41].

### 2.7. Immunofluorescence Staining

Immunostaining was performed following the protocol described in [42] on 5 µm paraffin sections of brain tissues. After deparaffinization, sections were saturated (2h at room temperature in TBS containing 0.2% triton, 0.5% FBS, and 1% BSA), then incubated with primary antibodies; anti-GFAP (Dako, Agilent, Santa Clara, USA, Z0334; 1:500) at 4 °C overnight, washed, and incubated with a secondary anti-rabbit Alexa 488 antibody (Abcam, ab150077, 1:1000). The slides were stained with DAPI for 10 min, washed with PBS, mounted with Fluoromount-G (SouthernBiotech, Birmingham, UK), and inspected visually using a ZEISS AXIOVERT 200 M Apotome microscope (Zeiss, Oberkochen, Germany) connected to a digital camera. Serial sections were analyzed at 200× magnification to reconstruct the whole hippocampus volume of the brain using the ZEN2.1 software (Zeiss). Images were collected as serial Z stack series from 18 optical slices. GFAP-positive cells were counted manually from *cornu ammonis* (CA) 1/CA2, CA3, and the dentate gyrus (DG) using Image J (version 1.53q, Fiji software) [43]. A total of 40 slides were analyzed, corresponding to treated group (*n* = 5) and control group (*n* = 5).

### 2.8. Primary Culture, FAβ_1–42_ Treatment, qRT-PCR, and ELISA

#### 2.8.1. Primary Astrocytes Preparation and Culture

Primary astrocyte cultures were prepared following the protocols described by Galland et al. [44]. Six brains of new-born Sprague–Dawley rats of 3 days of age were collected aseptically before to manually isolate the cerebral hemispheres. After being carefully removed, the meninges and tissues were dissociated mechanically and rinsed with PBS. The cell suspension was centrifuged at 1000× *g* for 5 min at 4 °C. Cell pellets were resuspended in DMEM complete medium and seeded in 24-well plates at density of 1.5 × 10^5^ cells/cm^2^ until they reached 80% confluency. The medium was changed every 3–4 days. The cells were thereafter maintained in tissue culture for approximately 15 days.

#### 2.8.2. Treatment of Primary Astrocytes with FAβ, LPS, BMS345541, and miRNA Inhibitor

Confluent cell monolayers were washed twice with PBS and then incubated with the FAβ_1–42_ solutions at final concentrations of 0.5, 1, and 2 µM for 3 days at 37 °C, 5% CO_2_. In parallel, cells were also treated with BMS-345541, a NF-κB inhibitor (Sigma-Aldrich, Saint-Quentin-Fallavier, France) used at a final concentration of 5 µM. When specified, the cells were pre-incubated for 1 h in culture with the BMS-345541 inhibitor before treatment with either the FAβ_1–42_ or LPS solution (100 ng/mL in complete media) for 3 days as described here [45,46]. Cells were also treated with commercially available synthetic miRNA inhibitors for miRNA-146a-5p (AMO-146a, MH10722, ThermoFisher Scientific, Waltham, MA, USA) or control (AMO-CTL, 4464076, ThermoFisher Scientific, Waltham, MA, USA) using the RNAimax transfection reageant (ThermoFisher Scientific) according to manufacturer’s recommendations. Briefly evoked, confluent cell monolayers were washed with PBS and then transfected with AMO-146a or AMO-CTL at 100 nM final concentration for 6 h in OPTIMEM medium (ThermoFisher Scientific). Then, the next day, cells were treated with FAβ_1–42_ or DMSO control solutions at indicated concentrations before being collected for tRNA extraction and qRT-PCR analysis.

#### 2.8.3. Cell Viability Assay

Cell viability analysis was performed with the Alamar Blue™ HS reagent according to manufacturer’s instructions (Invitrogen, Carlsbad, CA, USA, A50101). Briefly described, a 1/10 dilution of Alamar blue solution was added to each well of 24-wellplates for 2 h at 37 °C. Then, 50 µL of cell supernatant were monitored using a fluorescence microplate reader set up at 560 and 605 nm as excitation and emission wavelengths, respectively.

#### 2.8.4. miRNA and mRNA Quantification from Primary Astrocytes Culture

The procedure to quantify the relative miRNA expression from the primary astrocyte culture was the same as described above, except that no miRNA spike-ins were added to the samples and that the relative expression of the small nuclear RNA 6 (U6) was used to normalize the data as described before [47].The procedure to quantify the relative mRNA expression was also derived from routine procedures described by Simion et al. and Ezine et al. [38,39,47]. Briefly described, total RNA was extracted from cells using the Trizol reagent (Invitrogen, Carlsbad, CA, USA) and reverse transcribed from 100 ng tRNA using the RevertAid RT Reverse Transcription Kit from ThermoFisher (Thermofisher Scientific, Waltham, MA, USA). Commercially available primers (Qiagen, Hilden, Germany) were used to monitor expression of *Irak1*, *Traf6, Irak2*, *IL-6*, *IL-1β,* and *CXCL1* genes. The relative expression of *GAPDH* gene was used to normalize expression of mRNA transcripts [40].

#### 2.8.5. Sandwich Enzyme-Linked Immunosorbent Assay

A sandwich enzyme-linked immunosorbent assay (ELISA) system was used to quantify concentrations of IL-1β, IL-6, and CXCL1 from the culture medium of astrocytes, using antibodies paired according to routine procedure [48].

### 2.9. Statistical Analysis

All data were expressed as the mean ± SEM. Statistical analyses of the Morris test were performed with Stata 13.0 software, and the non-parametric Kruskal–Wallis test was used to evaluate the difference between the groups. To analyze the relative expression of miRNAs and mRNAs, the statistical software XLSTAT by Addinsoft was used, and the non-parametric Mann–Whitney U test was used to compare the expression of miRNAs between the groups. Experimental conditions within each in vitro experiment were performed in triplicate by a minimum of three independent experiments. A Student’s t-test was used to evaluate the difference in the relative expression of mRNA. All graphs were made using GraphPrism 8.0 software. Statistical significance was set at 95% confidence interval, with *p* values set as *** ≤ 0.05, ** ≤ 0.01, *** ≤ 0.001.

## 3. Results

### 3.1. Cognitive Impairment in Rats Inoculated with 1 µg/µL of FAβ_1–42_ Peptides Solutions

The rat model of AD, induced by the intrahippocampal infusion of FAβ_1–42_ peptide solutions, was first challenged using the classic Morris Water Maze test to evaluate the memory and behavior performance of treated animals [33]. For each rat, we monitored the distance traveled and the escape latency as variables of cognitive performance using a video-based system to automatically extract trajectories and the time used by rats to reach the platform on the maze.

During the acquisition session, no statistically significant difference was found at day 1 of the training session between all groups, indicating that animals from both groups have similar motor and visual abilities. In contrast, significant differences in path lengths were found from day 2 to day 5 (Figure 1A). The cognitive abilities were studied through the evaluation of performance in four quadrants. In the NW quadrant, path lengths showed that the FAβ_1–42_ treated rats used longer distances to find the platform over these 4 days of interval time compared with PBS-treated rats (Appendix A). When the rats were placed in the NE quadrant, significant differences were only observed at day 4 and day 5 (Appendix A). For rats placed in the SW quadrant, FAβ_1–42_-treated rats traveled longer distances compared with the control group of animals, but the differences were not statistically significant due to a great dispersion observed in this group (Appendix A). In the SE quadrant, the path lengths of the FAβ_1–42_-treated group were significantly different from those of the control group from day 2 to day 4 (Appendix A). The escape latency of each rat was evaluated in all four quadrants. The groups of animals infused with FAβ_1–42_ showed a longer escape latency compared with the control group, with a significant difference at day 4 and day 5 (*p* < 0.05) (Figure 1B), demonstrating a significant memory defect.

### 3.2. FAβ_1–42_ Infusion Leads to an Inflammatory Response in the Hippocampus of Rats

Astrogliosis is a well-recognized feature of AD, characterized by cellular hypertrophy and an increase in glial fibrillar acid (GFAP) expression [49]. To evaluate whether cognitive impairment detected in rats inoculated with FAβ_1–42_ peptide solution (1 µg/µL) was correlated with astrogliosis, we performed GFAP fluorescence labeling of brain tissues of rats harvested at 14 days post-infusion. Representative immunofluorescence images are shown in Figure 2A. Compared with brain sections from the control group, a more pronounced fluorescence staining was detected in the whole hippocampal tissues, including the CA1/CA2 and CA3 and DG regions of the brain sections from FAβ_1–42_-treated rats. The quantitative analysis of the whole fluorescence staining indicated that there were 2.1-fold and 1.7-fold more GFAP positive cells respectively in CA1/CA2 and CA3 and DG hippocampal regions of FAβ_1–42_-infused rats compared with control rats (Figure 2B,C).

### 3.3. Intrahippocampal Injection of FAβ_1–42_ Leads to Dysregulation of Circulating miRNAs

Then, we evaluated whether the defective cognitive performance and astrogliosis detected in FAβ_1–42_-treated rats might be correlated with dysregulation of a pattern of circulating miRNAs. We selected from the bibliography a list of nine miRNAs (miRNA-125b-5p, -181c-5p, -191-5p, -106b-5p, -135a-5p, -146a-5p, -9a-5p, -29a-3p, and -29c-3p) as these miRNAs are documented to be most commonly dysregulated in sera samples of animal models of AD and/or AD patients [25,26,27,28,29,31]. Results from qRT-PCR indicated that amongst this list, only miRNA-146a-5p was found to be statistically significantly different in sera of rats infused with FAβ_1–42_ peptide solution (1 µg/µL) compared with rats treated with PBS (Figure 3, *p* < 0.05). The average of expression of miRNA-9a-5p, -29a-3p, and -29c-3p in the FAβ_1–42_ -treated rats was lower compared with the control group, and very close to a statistical *p* value of 0.05 (Figure 3). In contrast, the relative expression of the other five miRNAs (miRNA-125b-5p, -181c-5p, -191-5p, -106b-5p, and -135a-5p) was far from significant.

### 3.4. Increasing Amount of Infused FAβ_1–42_ Peptides in Hippocampus of Rats Further Dysregulates the Relative Abundance of Circulating miRNAs in Serum Samples

We followed up our investigation by infusing 2.5-fold more FAβ_1–42_ peptide solution (e.g., 2.5 µg/µL) in the hippocampus of rats to evaluate whether the pattern of miRNA expression might be different. Data from qRT-PCR analysis made on serum samples collected at day 21 indicated that the expression of miRNA-146a-5p, -29a-3p, and -29c-3p were significantly dysregulated in the FAβ_1–42_ -treated group in this second cohort of animals. As for the first cohort, no significant change in the expression of miRNA-9a-5p was found. In contrast to the first cohort of rats infused with 1 µg/µL of FAβ_1–42_ peptides, miRNA-125b-5p and -191-5p were detected as the most dysregulated miRNAs in this second cohort of rats (Figure 4).

### 3.5. Kinetics of Circulating miRNA Expression Detected in Rats Infused with FAβ_1–42_ Peptides Solution Used at 2.5 µg/µL

Next, we postulated that the expression of these miRNAs might be differently dysregulated as a function of time post-infusion of the FAβ_1–42_ peptide solution. We included miRNA-9a-5p assuming that this miRNA might be dysregulated at earlier time points than at the 21-day post-infusion time used previously. Results of these kinetics are shown in Figure 5.

The kinetics of miRNA-146a-5p expression in the treated group was different compared with that of the control group. No statistically significant reduction of expression was found at the 7 day-early time point, followed by statistically significant reductions of expression detected at both the 14- and 21-day time points (*p* = 0.022 and *p* = 0.027, respectively, Figure 5A). The expression patterns of miRNA-29a-3p and -29c-3p were statistically reduced at all times compared with the control group, although the most significant difference was found at day 7 (*p* < 0.01; Figure 5B,C). Concerning miRNA-9-5p in FAβ_1–42_-infused rats, its kinetics of expression tends to decrease as a function of time compared with the control group but was not statistically different at each time point (Figure 5D).

### 3.6. Common miRNAs Dysregulated in APP_swe_/PS1_dE9_ Transgenic Mice Model and FAβ-Brain Infused Rat Model

Next, we sought to compare the expression pattern of circulating miRNAs detected in this FAβ_1–42_-infused animal model of AD with the widely used APP_swe_/PS1_dE9_ transgenic mice model of AD. This transgenic mice model expresses two constitutive mutant forms of APP and PSEN1 and produces Aβ_1–42_ deposits by 6 months of age, followed by abundant plaque apparition in the hippocampus and cortex by 9 months, which are increasing up to around 12 months of age [50].

We evaluated the expression of the above-mentioned miRNAs in serum samples of APP_swe_/PS1_dE9_ mice collected at 4 and 15 months of age. The results obtained indicated that circulating miRNAs detected in serum samples of 4-month-old APP_swe_/PS1_dE9_ mice were not significantly different from those of control littermate mice. In serum samples collected from 15-month-old APP_swe_/PS1_dE9_ transgenic mice, four miRNAs (miRNA-125b-5p, -29a-3p, -29c-3p, and -191-5p) were found to be significantly downregulated (Figure 6). Interestingly, these four miRNAs were also significantly downregulated in the cohort of animals infused with 2.5 µg/µL of FAβ_1–42_ peptide solution (Figure 4). In contrast, no statistically significant difference in terms of miRNA-146a-5p expression was detected in the sera of APP_swe_/PS1_dE9_ transgenic mice (*p* > 0.179) independent of their age, whereas it was found to be statistically significantly downregulated in the FAβ_1–42_-infused rats (*p* < 0.027). Moreover, miRNA-125b-5p was found statistically much more downregulated (**** p* < 0.0004) in the FAβ-infused rats’ model of AD as compared with the APP_swe_/PS1_dE9_ transgenic mice (*p* < 0.02). The same could be stated for miRNA-181c-5p, which, although not statistically significantly dysregulated in the two animal models of AD, had a *p*-value close to significance (*p* < 0.070) in the rat model as compared with APPswe/PS1dE9 transgenic mice (*p* < 0.748).

### 3.7. Intrahippocampal Inoculation of FAβ_1–42_ Leads to Increased Expression of miR-146a in CSF

We were intrigued by the observation that the kinetics of miRNA-146a-5p expression detected in the FAβ_1-42-_infused animal model of AD were different from other miRNA kinetics and the absence of dysregulation found in APPswe/PS1dE9 mice. This prompted us to focus our next experiments on deciphering the biological impact of miR-146a induction in the pathogenesis of the FAβ_1–42_-infused animal model of AD.

First, we evaluated the expression pattern of miRNA-146a-5p in the CSF sample collected 14 days after the intrahippocampal infusion of FAβ_1–42_ peptide solution (2.5 μg/μL). Results indicated that, in contrast to serum samples (Figure 4; *p* = 0.022), the amount of miRNA-146a-5p in CSF samples was detected as statistically significantly up-regulated (Figure 7; *p* = 0.004). This data highlights the discrepancy between the presence of miRNA-146a-5p in serum versus CSF samples.

### 3.8. FAβ_1–42_ Peptides Induces the Expression of miRNA-146a-5p in Primary Astrocytes through the NF-κB Cell Signaling Pathway but Without Inducing Production of Pro-Inflammatory Cytokines

To gain insight into the biological function of miRNA-146a-5p in the pathogenesis of the FAβ-induced animal model of AD, we performed a mechanistic study with a primary culture of rat astrocytes treated with the FAβ_1–42_ peptide solution. Our objective was to recapitulate, at least partially, the AD-like environment induced by the infusion of FAβ_1–42_ peptides in the hippocampi of rats. Astrocytes are considered to promote the first line of the neuroinflammation response [51] by regulating the expression of key mediators of innate and adaptive immune responses in the central nervous system, which are responsible in part for the early onset of cognitive dysfunction.

First, we searched for the minimal concentration of FAβ_1–42_ peptide solution that did not induce cytotoxicity in astrocytes to mimic early biological events associated with the presence of FAβ_1–42_ peptides in the hippocampi of animals. Results indicated that FAβ_1–42_ peptide solutions ranging from 0.5 to 2 µM were well tolerated by these cells, whereas at higher concentrations, significant toxicities were observed (Appendix A). Next, we monitored miRNA-146a-5p expression in these cells after treatment with 0.5, 1, and 2 µM of FAβ_1–42_ solutions. The qPCR data demonstrated that miRNA-146a-5p expression tended to increase with the FAβ_1–42_ peptide concentration used, with a maximum of 1.5-fold induction of miRNA-146a-5p expression detected when cells were incubated with 2 µM of FAβ_1–42_.

The expression of miRNA-146a-5p has been reported to be upregulated in several central nervous cells in response to TNF-α, IL-1β, or LPS through the activation of the NF-κB cell signaling pathway [52,53,54]. In the next step, we evaluated whether FAβ_1–42_ treatment might induce the expression of miRNA-146a-5p in primary astrocytes using the same cell signaling pathway. For that, cells were treated for 3 days with 2 µM of FAβ_1–42_ peptide solution in the presence or absence of BMS-345541, a pharmacological inhibitor of the NF-κB pathway, and then miRNA-146a-5p expression was evaluated by qRT-PCR. As a positive control, cells were treated with LPS [55]. Results shown in Figure 8 confirm that miRNA-146a-5p expression was significantly upregulated in FAβ_1–42_-treated cells as compared with PBS-treated cells used as controls. This induction was dependent on NF-κB cell signaling. Indeed, pre-treatment of cells with BMS-345541 inhibitor prior to the incubation with FAβ_1–42_ peptide solution significantly dropped the expression level of miRNA-146a-5p to the basal expression level detected in non-treated cells (Figure 8B). As expected, LPS-treatment of primary astrocytes increased significantly the expression of miRNA-146a-5p, which can be significantly reversed by BMS-345541 inhibitor treatment. Those results indicate that FAβ_1–42_ peptide treatment increases the basal expression level of miRNA-146a-5p through the transcriptional regulation of the NF-κB pathway, as previously demonstrated with pro-inflammatory cytokines [53,54] and here with LPS.

We next evaluated the expression of IRAK-1/2 and TRAF-6, three well-known transcriptional targets of miRNA-146a-5p involved in the TLR4/NF-κB cell signaling pathway [52,56]. As shown in Figure 8C, the relative expression of IRAK-1 was significantly downregulated in FAβ_1–42_-treated cells compared with control cells, while the expression of IRAK-2 was unchanged. Surprisingly, the relative expression of TRAF-6, a direct downstream effector of the IRAK-1/2 complex, was unchanged in these FAβ_1–42_-treated cells (Figure 8C).

Based on the above results, we next checked whether the downregulation of IRAK-1 expression detected in astrocytes treated with FAβ_1–42_ peptides might be sufficient to induce the expression of pro-inflammatory cytokines. As shown in Figure 9, none of the pro-inflammatory cytokines evaluated (IL-6, IL-1β, and TNF-α) were induced in FAβ_1–42_-treated cells (Figure 9A), as was the case for CXCL1. In contrast, significant and high induction levels of IL-6; IL-1β and CXCL1 were detected in LPS-treated cells used as positive controls (Figure 9B).

### 3.9. Anti-miRNA-146a-5p Oligonucleotides (AMO-146a) Rescue the Expression of IRAK-1 and TRAF-6 as Well as Pro-Inflammatory Cytokines in FAβ_1–42_-Treated Astrocytes

We hypothesized that interfering with the expression of miRNA-146a-5p in astrocytes might rescue IRAK-1 and TRAF-6 expression and then promote cytokine production in response to FAβ_1–42_ treatment.

At first, we evaluated the performance of an anti-miRNA-146a-5p oligonucleotide (AMO-146a) to inhibit the expression of miRNA-146a-5p in astrocytes. As shown in Figure 10A, the transfection of AMO-146a in these cells led to a statistically significant down-regulation (*p* = 0.01) of miRNA-146a-5p expression, which was not observed with AMO control (AMO-CTL). Then, we evaluated the biological impact of AMO-146a on the expression of IRAK-1/2 and TRAF-6 as transcriptional targets of miRNA-146a-5p. Cells were first transfected with AMO-146a or AMO-CTL, then treated with FAβ_1–42_ peptides, and finally lysed 48 h later to quantify the relative expression of IRAK-1/2 and TRAF-6. As shown in Figure 10B, AMO-146a delivery significantly elevated the expression of IRAK-1 (*p* = 0.042) and TRAF-6 (*p* = 0.0035), while AMO-CTL did not change the expression of these transcriptional targets. In contrast, IRAK-2 expression remained unchanged following the transfection with both AMO-146a and AMO-CTL. Moreover, it should be noted that AMO-146a treatment led to increased IRAK-1 and TRAF-6 expression levels superior to those detected in non-treated cells (NT). These last results suggest that interfering with miRNA-146a-5p expression rescues not only the expression of IRAK-1 and TRAF-6 in FAβ_1–42_-astrocyte cells but also induces a higher expression of those targets in non-treated astrocytes (NT).

Next, we evaluated whether rescuing expression of IRAK-1 and TRAF-6 by AMO-146a will be sufficient to induce the expression of cytokines in FAβ_1–42_-treated astrocytes. As shown in Figure 11, the transfection of AMO-146a before treatment with FAβ_1–42_ peptides resulted in a statistically significant induction of IL-6 (*p* = 0.003), IL-1β (*p* = 0.05), and CXCL1 (*p* = 0.01) but not TNF-α (*p* = 0.803). The transfection of AMO-CTL had no impact on the expression of these molecules.

## 4. Discussion

In this study, we sought to identify a panel of miRNAs that might be directly correlated with the production of Aβ in adult tissues of the brain of animals as an early diagnosis marker. Accumulation of Aβ in the brain is one of the first steps of AD pathogenesis that ultimately causes inflammation and cognitive dysfunction. Therefore, the injection of fibrillar forms of Aβ in the hippocampus of an adult animal could be used as a model to investigate the very early induced events of AD. To the best of our knowledge, no reports have described a panel of several circulating miRNAs in serum samples of adult rats infused with FAβ_1–42_ peptides into the hippocampus, and none of them have compared the pattern of miRNA expression that could be detected in the APP_swe_/PS1_dE9_ transgenic mice model of AD. To address this point, we selected a short list of nine circulating miRNAs frequently dysregulated in patients with AD and/or in transgenic animal models and performed a comparative study between these two animal models of AD. We found a striking difference in terms of the expression pattern of some of these miRNAs and demonstrated that miRNA-146a-5p was exclusively dysregulated in the FAβ_1–42_-infused rat model of AD. We then provided experimental evidence that this miRNA was enriched in the cerebrospinal fluid of FAβ_1–42_-infused adult rats and that its expression was induced in primary astrocytes following treatment with FAβ_1–42_ peptides. Data from loss-of-function studies combined with pharmacological inhibition of the NF-κB pathway suggest that this miRNA acts as an anti-inflammatory modulator in astrocytes through a negative feedback loop of the NF-κB pathway, impairing cytokines and chemokines production. Taken together, our data indicate that miRNA-146a-5p might be considered an early blood-circulating miRNA marker reflecting the presence of an aggregated form of Aβ in adult brain tissues, in which it plays anti-inflammatory roles.

The hippocampal infusion of Aβ peptides has been shown to reduce neuronal density, increase expression of the glial fibrillar acid protein, and cause deficiencies in the behavioral performance of infused animals [57]. Borbely et al. demonstrated that intrahippocampal administration of synthetic Aβ peptides simultaneously decreases both the spatial learning capacity in MWM and the density of the dendritic column in the CA1 region of the rat hippocampus [30]. We did observe the same following the infusion of FAβ_1–42_ peptides into the CA1 region of the hippocampus and observed a deterioration of the learning capacity of these animals, mainly at day 14 post-surgery [58]. Abnormal accumulation and shedding of Aβ peptides can lead to localized inflammation involving reactive astrocytes with increased expression of GFAP [49]. This gliosis process occurs after brain injury and is characteristic of neurodegenerative disorders such as AD [59]. We validated the presence of reactive astrocytes by immunofluorescence staining of GFAP expression in brain sections of animals infused with FAβ_1–42_. They had a higher number of astrocytes in all regions of the hippocampus (CA1/CA2, CA3, and DG), suggesting a reactive state. Each area contained approximately twice the number of detectable astrocytes than that of control animals (Figure 2). These data are in line with those of different studies showing an increase in astrocytes in a model of injection of fibrillar Aβ in the hippocampus of a rat model of AD [57,60].

Since miRNAs are known to act as temporal regulators of different biological processes, we established the kinetics of the expression of miRNA-9-5p, -29a-3p, -29c-3p, and -146a-5p [61,62] at days 7, 14, and 21 following intrahippocampal infusion of FAβ_1–42_ in adult rats (Figure 5). Kinetic studies of those miRNAs revealed a clear downregulation tendency, although no statistically significant difference was detected for miR-9-5p expression. Members of the miRNA-29 family showed a strong dysregulation in the FAβ_1–42_-injected group at 7 days post-injection compared with the control group. At 14 and 21 days, the dysregulation was maintained, but the curve started to rise very mildly, suggesting a reverse phase likely resulting from the clearance of FAβ_1–42_ peptides by macrophages and/or microglia/astrocytes (Figure 5). The downregulation of miRNA-29a-3p and miRNA-29c-3p is consistent with many previous reports that have evaluated serum and/or plasma from patients with AD [31,63,64]. Interestingly, miRNA-29 a/b has been described as a regulator of BACE1/beta-secretase expression [29,65] through binding to targeting moieties located in the 3′UTR of BACE1 mRNA. The same observation can also be made for miRNA-191-5p, which is frequently downregulated in the sera of AD patients [16] as well as in our FAβ_1–42_-infused animal model. Assuming that BACE1/beta-secretase expression contributes to Aβ accumulation in the brain of other animal models of AD [66,67], it might be possible that lowering the expression of miRNA-29 a/b-3p and -191-5p in FAβ_1–42_-infused rats might result in an increase of BACE1 expression and consequently Aβ_1–42_ peptide production by neural cells. This process might directly participate in or even amplify the astrogliosis process revealed in our study by GFAP staining. The expression of miRNA-146a-5p was also downregulated in FAβ_1–42_-infused rats compared with the control group, even though the reduction was only statistically significant after 14 and 21 days. MiRNA-146a-5p has been shown to be an early and prevalent pathological biomarker of AD as it is involved in the inflammatory response and neuroinflammation [25,68]. Our results are in line with different studies that have evaluated the circulating expression profile of miRNA-146a-5p in AD animal models and in humans, although its temporal expression over the pathogenesis of AD was not reported [25,68]. Mechanistically, miRNA-146a-5p acts as a negative regulator of NF-κB signaling to prevent the overproduction of pro-inflammatory cytokines or chemokines [53,56,69]. Therefore, the peak of miRNA-146a-5p dysregulation detected at day 14 in the FAβ_1–42_-infused animal model could be correlated with the peak of an inflammatory reaction induced by the presence of the FAβ_1–42_ peptide, for which miRNA-146a-5p might exert its maximal anti-inflammatory function. Surprisingly, no significant difference in the expression of miRNA-146a-5p was detected in the serum samples of APPswe/PS1dE9 transgenic mice, although these mice develop Aβ deposits [70]. This discrepancy between these two animal models of AD can be explained by the fact that the FAβ_1–42_ infusion model represents a model of inflammation induced by the direct exposition of hippocampal tissues to FAβ_1–42_ peptide loads [71], while the APPswe/PS1dE9 transgenic model corresponds to a model of chronic inflammation induced by the gradual accumulation of Aβ deposits over the lifespan of transgenic animals, which express constitutive mutant forms of APP and PS1 [72]. Therefore, the inflammatory responses are expected to be different in these two models and should implicate different cellular pathways and molecular partners. This difference might also be accentuated by compensatory or adaptive responses developed in Knock-OUT or Knock-IN transgenic animals [73,74] that counteract the biological impact of mutants. To complement our data, we quantified the amount of miRNA-146a-5p in CSF obtained from FAβ_1–42_-infused rats at 14 days post-injection. The expression of miRNA-146a-5p was detected as significantly upregulated in contrast to the serum sample. Reasons for this difference are not clear and difficult to apprehend. The same trend of opposite regulation between sera samples and CSF was described for other miRNAs [16].

Neuroinflammation is one of the hallmarks of AD, and miRNA-146a-5p might be a key mediator of the immune response linked to a variety of inflammation processes. To understand further the role of miRNA-146a-5p, we conducted functional in vitro studies with primary rat astrocytes, the most abundant cell type in the CNS and important modulators of the neuronal innate and inflammatory immune responses [75]. We observed a statistically significant upregulation of miRNA-146a-5p expression in astrocytes treated with FAβ peptides compared with control cells (Figure 8). These findings are consistent with studies done on human astrocytes that had an upregulation of miRNA-146a-5p when exposed to Aβ peptides [51,53]. Similarly, Li et al. demonstrated that miRNA-146a-5p was positively regulated in human neuronal-glial (HNG), human astroglial (HAG), and human microglial (HMG) cells treated with Aβ_1–42_ peptides [53]. However, neuronal cells or astrocytes were cultured under oxidative stress or inflammatory conditions (H_2_O_2_, LPS, TNF-α) in addition to Aβ treatment in these studies. The expression of miRNA-146a-5p occurred through activation of the NF-κB pathway, which in turn induced the production of inflammation and cytokines. Since there are several NF-κB-binding sites in the promoter sequence analysis of miRNA-146a-5p, the direct effect of FAβ_1–42_ peptides on the expression of miRNA-146a-5p was not independently investigated in these studies [56]. We did also find that the expression of miRNA-146a-5p was dependent on activation of the NF-κB pathway, as pre-treatment of FAβ_1–42_-treated cells with the BMS-345541 pharmacological inhibitor of NF-κB significantly reduced miRNA-146a-5p expression. However, we did not detect the induction of IL-6, IL-1β, and TNF-α inflammatory cytokine production. This indicates that the production of cytokines in cells treated with Aβ in the presence of H_2_O_2_, LPS, or TNF-α is more likely driven by these inflammatory mediators than by Aβ peptides itself. To go further into the dissection of underlying molecular mechanisms governing expression of miRNA-146a-5p in FAβ_1–42_-treated cells, we quantified expression of downstream modulators of the NF-κB pathway as IRAK-1/2 and TRAF-6 that harbor miRNA-146a-5p binding sites in the 3′UTR mRNAs. These three genes encode key adaptor molecules downstream of Toll-like and cytokine receptors. Different reports [53,76,77] made with human astrocytes or nerve cells showed that, concurrent to miRNA-146a-5p upregulation in cells exposed to Aβ peptides and inflammatory molecules, a decrease of IRAK-1 associated with a compensatory increase in the expression of IRAK-2 was observed, as well as dysregulation of TRAF-6 [69,78]. Consistent with this observation, we found that in FAβ-treated cells, while IRAK-1 expression was downregulated whilst IRAK-2 and TRAF-6 expression was unchanged. TRAF-6 is described as one of the final effectors of NF-κB signaling, regulating the nuclear processing of IKKB for transcription of pro-inflammatory cytokines [79]. This latest result could explain the lack of cytokine production in FAβ-treated astrocytes. Therefore, it is tempting to believe that the treatment of astrocytes with FAβ_1–42_ peptides alone is sufficient to induce the transcription of the miRNA-146a-5p gene, which in turn might bind to IRAK-1 and, to lesser extent, IRAK-2 and TRAF-6. As a consequence of this retro-control loop of the NF-κB pathway, an abrogation of the expression of pro-inflammatory cytokines such as IL-6, TNF-α, and IL-1β might occur by keeping constant the expression level of TRAF-6. In contrast, in overactivated cells treated with FAβ_1–42_ peptides *plus* proinflammatory molecules, as reported by others [52,53,54], this retro-control loop of the NF-κB pathway by miRNA-146a-5p might be overridden, leading to a significant change in TRAF-6 expression and consequently the production of pro-inflammatory cytokines. This is in agreement with reports showing that miRNA expression can act as a negative feedback regulator of the same signaling pathway used for its own induction, thereby preventing an overstimulation of the inflammatory response [80]. To verify further this hypothesis, we transfected astrocyte cells with a miRNA-146a-5p inhibitor (AMO-146) prior to stimulation with FAβ peptide solution (Figure 11). As expected, a significant upregulation of IRAK-1 mRNA, a transcriptional target of the NF-κB pathway, was detected, whereas no change in IRAK-2 expression was found. In addition, the expression level of TRAF-6 was significantly upregulated. Remarkably, these changes in IRAK-1 and TRAF-6 expression were directly correlated with the production of de novo pro-inflammatory cytokines such as IL-6, IL-1β, and CXCL1. Therefore, the increased level of TRAF-6 expression induced by AMO-146 allows it to reach a threshold that might lead to an activation of upstream molecular mediators of NF-κB signaling and, as consequence, the production of inflammatory cytokines. Additional studies are required to elucidate in detail the exact molecular mechanisms by which TRAF-6 overexpression led to the activation of the NF-κB pathway. We do not exclude the existance or participation of other cell signaling pathways. Nevertheless, the observation that interfering with miRNA-146a-5p expression is sufficient to modulate the production of inflammatory cytokines in FAβ_1–42_-treated astrocytes tends to support our overall statement. In suboptimal conditions, e.g., the presence of FAβ_1–42_ peptides alone, mimicking the early stage of Aβ deposit, miRNA-146a-5p might play an anti-inflammatory role by down-regulating expression of IRAK-1, resulting in the maintenance of the steady state of TRAF-6 expression and consequently the inhibition of inflammatory cytokine production. On the contrary, in advanced stages of AD pathology characterized by hyperproduction of pro-inflammatory cytokines in response to Tau phosphorylation and alteration of ApoE expression, for instance, the anti-inflammatory function of miRNA-146a-5p might be countered by multiple upstream signaling events activated by these inflammatory mediators of AD pathology. As a consequence, the expression of miRNA-146a-5p might not be sufficient to maintain neural tissues in a non-inflammatory state. Although deeper investigations are required to validate this statement, overall, our data provide additional insights into the molecular mechanism of anti-inflammatory function of miRNA-146a-5p and support the potential therapeutic function of miRNA-146a-5p in the management of AD as described by Mai et al. [81].

## 5. Conclusions

Data from this study show that circulating miRNA-146a-5p, -29a-3p, -29c-3p, -191-5p, and -125b-5p in the sera of rats injected with Aβ aggregates inside the hippocampus area were downregulated compared with those of control rats. Interestingly, some of those circulating miRNAs except miRNA-146a-5p were also dysregulated in the widely used APP_swe_/PS1_dE9_ transgenic mice model. By contrast, miRNA-146a-5p was upregulated in the CSF of those rats that presented astrogliosis in their brain. The mechanistic study done on rat primary astrocytes revealed that their treatment with Aβ aggregates also led to the upregulation of miRNA-146a-5p via the NF-κB signaling pathway, which in turn downregulated the expression of IRAK-1 but without affecting the expression of TRAF-6, a key effector of this NF-κB signaling pathway. As a consequence, no change in the expression of IL-6, IL-1β, and TNF-α was detected. A loss-of-function study performed with a miRNA-146a-5p inhibitor reversed this process and led to the production of IL-6 and IL-1β cytokines, as well as CXCL1. Based on those data and those reported in the literature, we propose that miRNA-146a-5p upregulation in astrocytes is likely playing an anti-inflammatory role through a negative feedback of the NF-κB pathway. In summary, this study contributes to improving our knowledge of the rat model of AD generated by intrahippocampal injection of Aβ_1–42_ peptides. As this model is supposed to characterize the very early stages of AD, the pattern of dysregulated miRNAs that we reported in this study deserves additional investigation for potential early diagnostic purposes.

## Figures and Tables

**Figure 1 cells-12-00694-f001:**
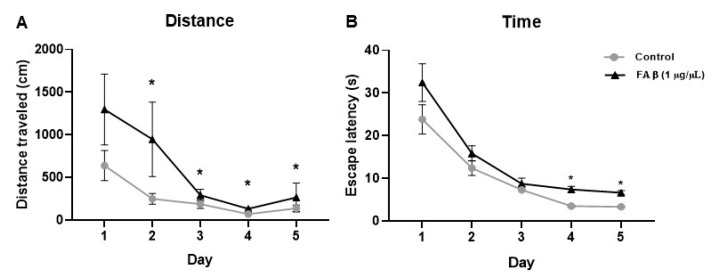
Effects of FAβ_1–42_ infusion on learning and memory capacity evaluated by the Morris Water Maze (MWM) test. The distance and the escape latency performed by rats (*n* = 10) injected with FAβ_1–42_ at a concentration of 1 µg/µL was evaluated at day 14 post-injection. (**A**) Total distance traveled and (**B**) Total escape latency used by rats during the 5 days of training. Control group (*n* = 10) of animals was injected with PBS. Data are represented as mean ± SEM. Statistical comparisons were performed using Kruskal–Wallis. ** p* ≤ 0.05.

**Figure 2 cells-12-00694-f002:**
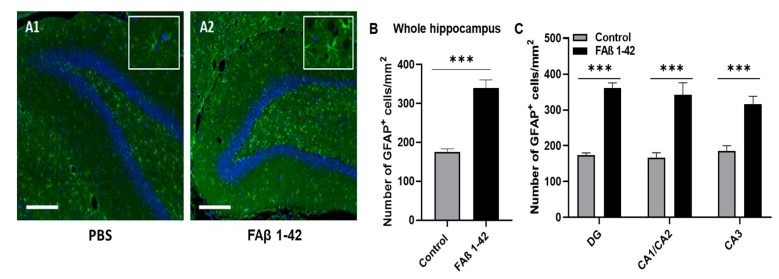
Quantification of GFAP^+^ astrocytes in the hippocampal areas. Brain sections from rats infused with FAβ_1–42_ (1 µg/µL) or PBS solutions were analyzed 14 days post-injection. (**A1**,**A2**) Representative immunohistochemical staining of GFAP expression (in green) in the DG area. The slides were counterstained with DAPI (blue). Scale bar, 500 µm. The box corresponds to GFAP staining with a higher magnification (**B**) Histograms showing the number of GFAP + cells per mm^2^ quantified in the total hippocampus and (**C**) in each area of the hippocampus (CA1/CA2, CA3 and DG). The quantification of GFAP + cells was analyzed in 4–5 sections per animal (*n* = 5 for each group). Statistical comparisons between both groups were analyzed using a Student’s *t*-test. **** p* ≤ 0.001.

**Figure 3 cells-12-00694-f003:**
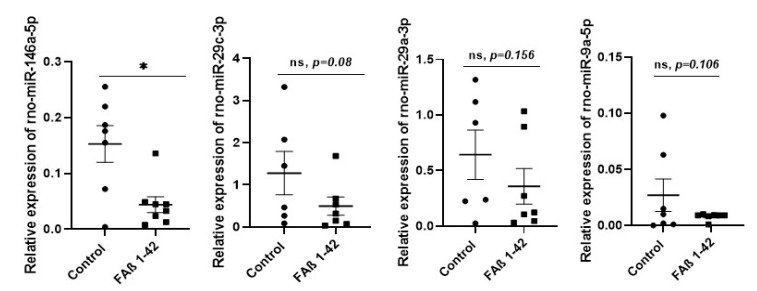
Relative expression of circulating miRNA-146a-5p, miRNA-29a-3p, miRNA-9a-5p, miRNA-29c-3p in serum samples of rats infused with 1 µg/µL of FAβ_1–42_ peptides solution. Sera from rats infused with FAβ_1–42_ peptides (treated group) or PBS (control group) solutions were collected 21 days post-infusion. MiRNA were extracted and quantified by qRT-PCR. Data are represented as mean ± SEM of 7–8 samples evaluated in triplicate. Statistical comparisons between FAβ_1–42_ and PBS group were performed using the Mann–Whitney test ** p* ≤ 0.05.

**Figure 4 cells-12-00694-f004:**
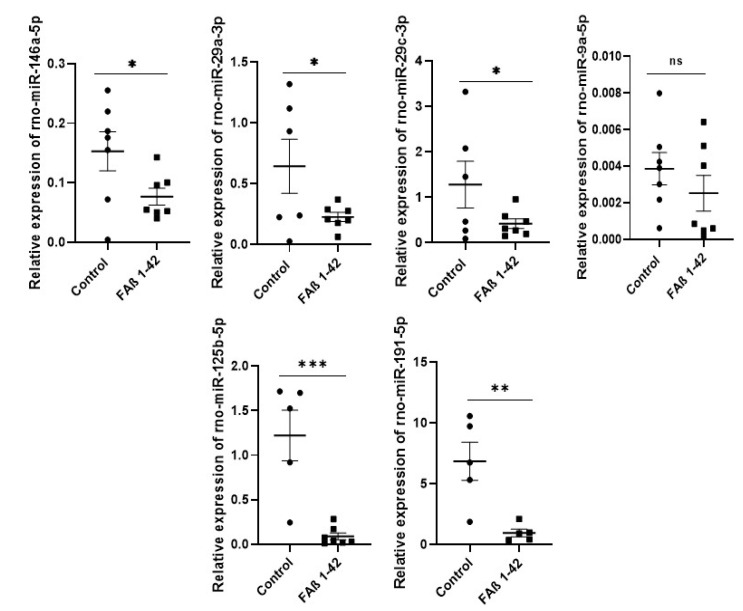
Relative expression of circulating miRNA-146a-5p, miRNA-29a-3p, miRNA-9a-5p, miRNA-29c-3p, miRNA-125b, miRNA-191-5p in serum of rats infused with 2.5 µg/µL of FAβ_1–42_ peptide solution. Serum samples were collected 21 days post-infusion. MiRNAs were extracted and quantified by qRT-PCR. Data are represented as mean ± SEM of 7–8 samples evaluated in triplicate. Statistical comparisons between FAβ_1–42_ and PBS group were performed using the Mann–Whitney test. * *p* ≤ 0.05, ** *p* ≤ 0.01, *** *p* ≤ 0.001.

**Figure 5 cells-12-00694-f005:**
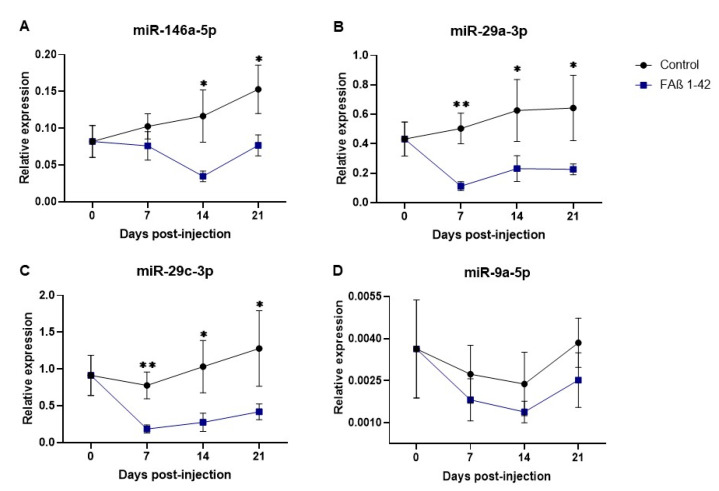
Kinetics of circulating microRNAs detected in the serum of rats infused with FAβ _1–42_ peptide solution used at 2.5 µg/µL final concentration. The quantification of miRNAs in serum of rats was investigated at 0, 7, 14, and 21 days post-infusion with FAβ_1–42_ peptides or with PBS control solutions. Relative expression profiles of (**A**) miR-146a-5p, (**B**) miR-29a-3p, (**C**) miR-29c-3p, and (**D**) miR-9a-5p quantified by qRT-PCR performed in triplicates and expressed as mean ± SEM (*n* = 8 rats per group and for each time point). Statistical comparisons between groups at each time were made using the Mann–Whitney test. ** p* ≤ 0.05, ** *p* ≤ 0.01.

**Figure 6 cells-12-00694-f006:**
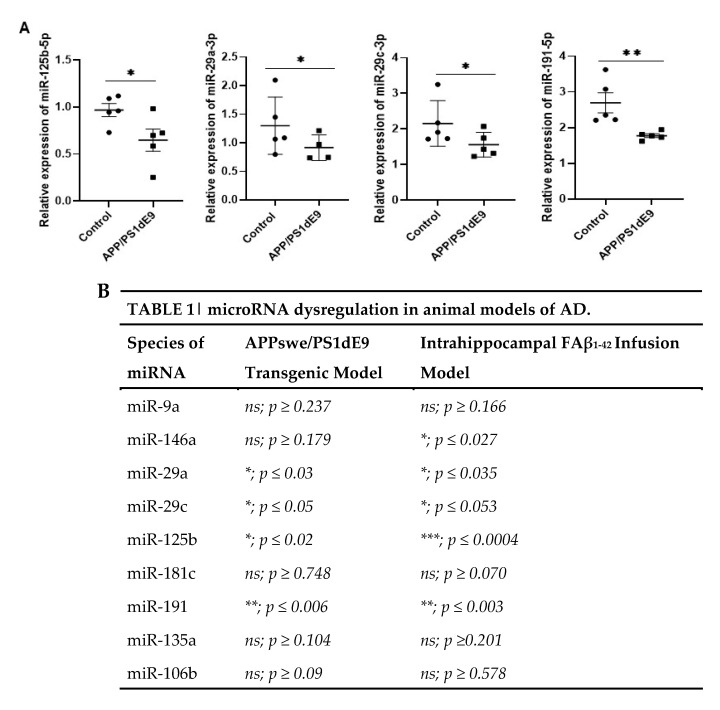
Relative expression of circulating miRNA-125b-5p, miRNA-29a, miRNA-29c, and miRNA-191-5p in serum of APP_swe_/PS1_dE9_ transgenic animal model of AD. (**A**) Serum samples from transgenic mice at 15 months of age were extracted and processed for miRNA detection by qRT-PCR. Data are represented as mean ± SEM of 5 sera samples performed in triplicate. (**B**) Table illustrating the statistically significant differences in terms of relative expression of miRNAs between the FAβ_1–42_-infused animal model (day 21) and the APPswe/PS1dE9 transgenic animal model of AD. Statistical comparisons between the transgenic and control group were performed using the Mann–Whitney test. ** p* ≤ 0.05, *** p* ≤ 0.01, *** *p* ≤ 0.001.

**Figure 7 cells-12-00694-f007:**
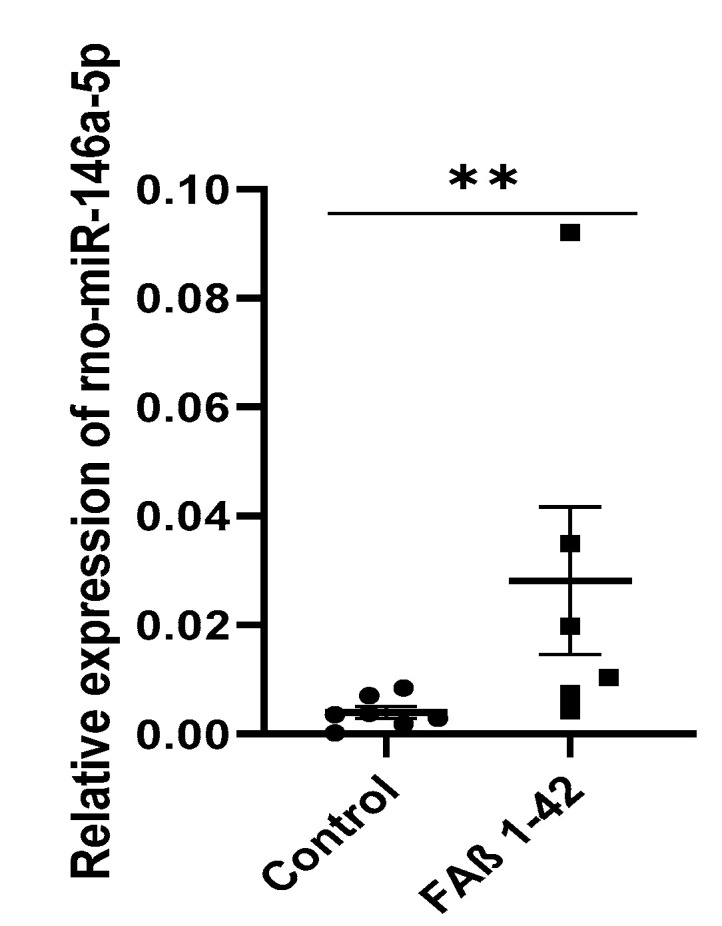
Relative expression of circulating miRNA-146a-5p in CSF of rats infused with FAβ_1–42_ peptides solution. CSF samples from rats infused with FAβ (2.5 µg/µL) or PBS solutions were collected at 14 days post-injection. The amount of miR-146-5p was then quantified by qRT-PCR. Data are represented as mean ± SEM of *n* = 7 samples performed in triplicate. Statistical comparisons between the FAβ-infused and control group were performed using the Mann–Whitney test. *** p* ≤ 0.01.

**Figure 8 cells-12-00694-f008:**
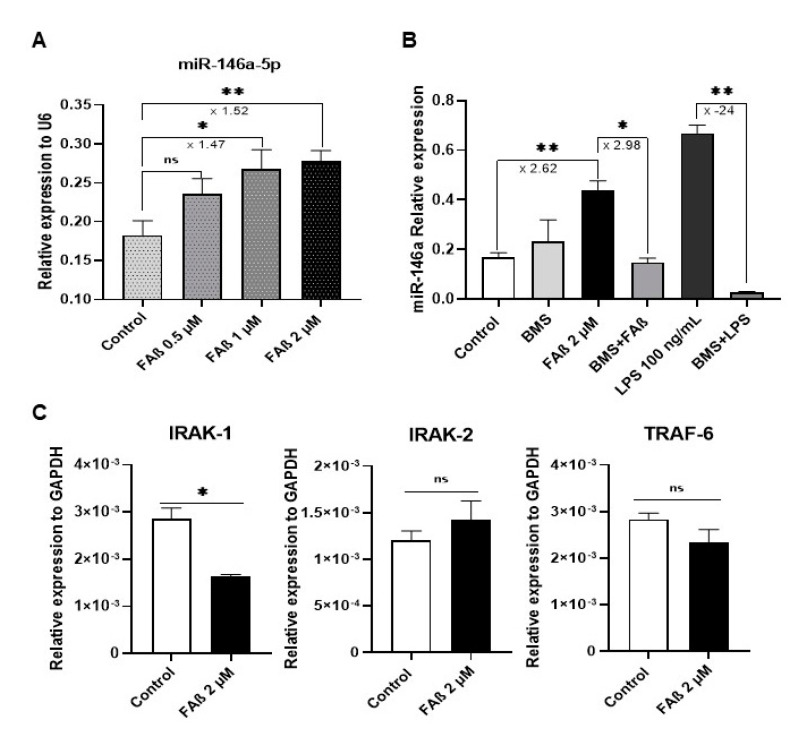
MiRNA-146a-5p is up-regulated in primary astrocytes treated with FAβ_1–42_ peptides, and its expression is dependent on the NF-κB pathway. (**A**) Relative expression of miRNA-146a-5p in primary astrocytes was 0.5, 1, and 2 µM of FAβ_1–42_ peptides for 3 days in tissue culture. (**B**) Relative expression of miRNA-146a-5p in primary astrocytes as function of treatment with 2 µM of FAβ_1–42_ solution in the presence of 5 µM of BMS-345541, IκB kinase pharmacology inhibitor. As positive control, cells were treated with LPS (100 ng/mL). (**C**) Relative expression of IRAK-1/2 and TRAF-6 in astrocytes treated with 2 µM of FAβ_1–42_ peptides. Data are represented as the mean ± SEM performed in triplicate. Statistical comparisons between FAβ_1–42_-treated cells and control, DMSO-treated cells were performed with a Student’s *t*-test. ** p* ≤ 0.05, ** *p* ≤ 0.01.

**Figure 9 cells-12-00694-f009:**
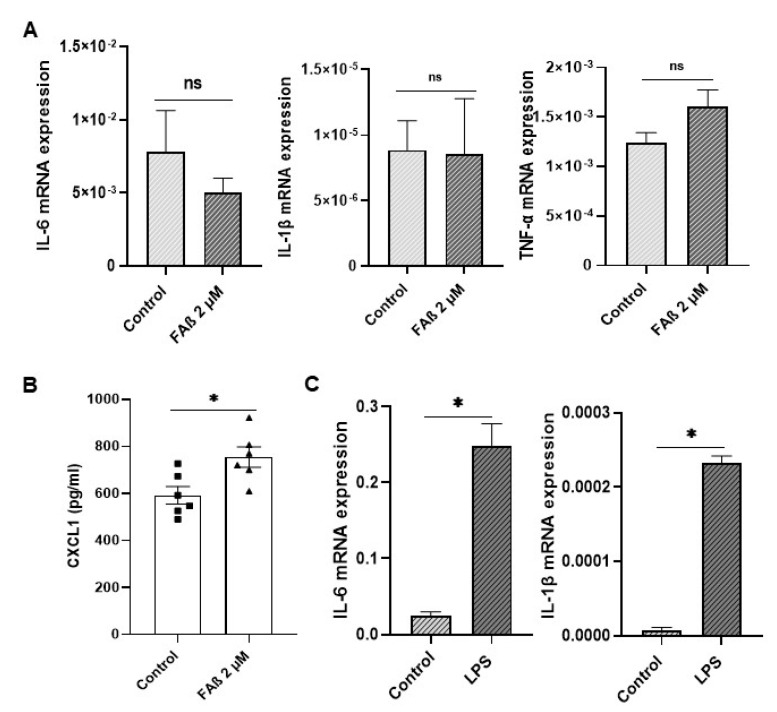
Expression of inflammatory markers in astrocyte cells treated with FAβ_1–42_ peptides solution. (**A**) Three days post-stimulation with FAβ solution (2 µM), the expression of IL-6, IL-1β, and TNF-α were quantified by RT-qPCR. (**B**) CXCL1 was quantified by ELISA. (**C**) Astrocytes were also stimulated with LPS at 100 ng/mL for 3 days as positive control. Data are represented as mean ± SEM of experiments made in triplicate. Statistical comparisons between groups were made using a Student’s *t*-test. ** p* ≤ 0.05.

**Figure 10 cells-12-00694-f010:**
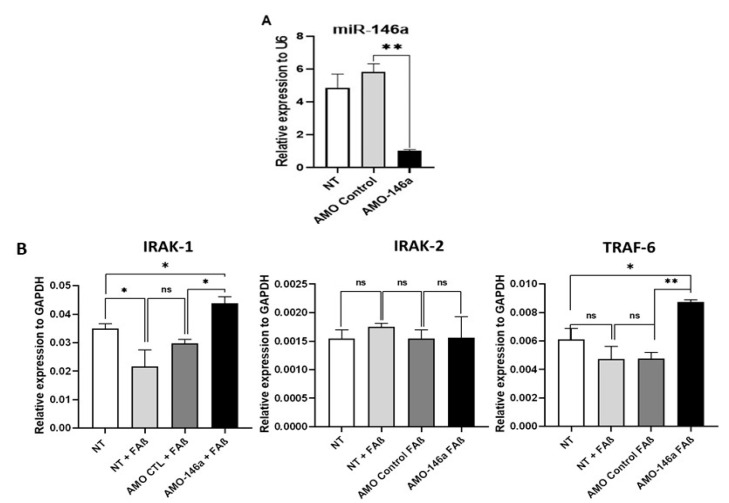
Transfection of miRNA-146a-5p inhibitor in FAβ_1–42_-treated astrocyte cells rescues expression of IRAK-1 and TRAF-6 but not IRAK-2. (**A**) qRT-PCR evaluation of performance of anti-miR-146a oligonucleotides (AMO-146a) to down-regulate expression of miRNA-146a-5p in astrocytes. (**B**) qRT-PCR evaluation of impact of AMO-146a transfection on expression of transcriptional targets of miRNA-146a-5p in FAβ_1–42_-astrocyte-treated cells. Cells were first transfected with AMO-146a or AMO-CTL (100 nM), then treated with FAβ_1–42_ peptides solutions (2 µM) before quantification of the relative expression of IRAK1/2 and TRAF-6 transcriptional targets of miRNA-146a-5p. Non-treated astrocytes (NT, no FAβ_1–42_ treatment) were used as additional control. Data are represented as mean ± SEM of one representative experiment performed twice in triplicate. Statistical comparisons between groups were made using a Student’s *t*-test. ** p* ≤ 0.05, *** p* ≤ 0.01.

**Figure 11 cells-12-00694-f011:**
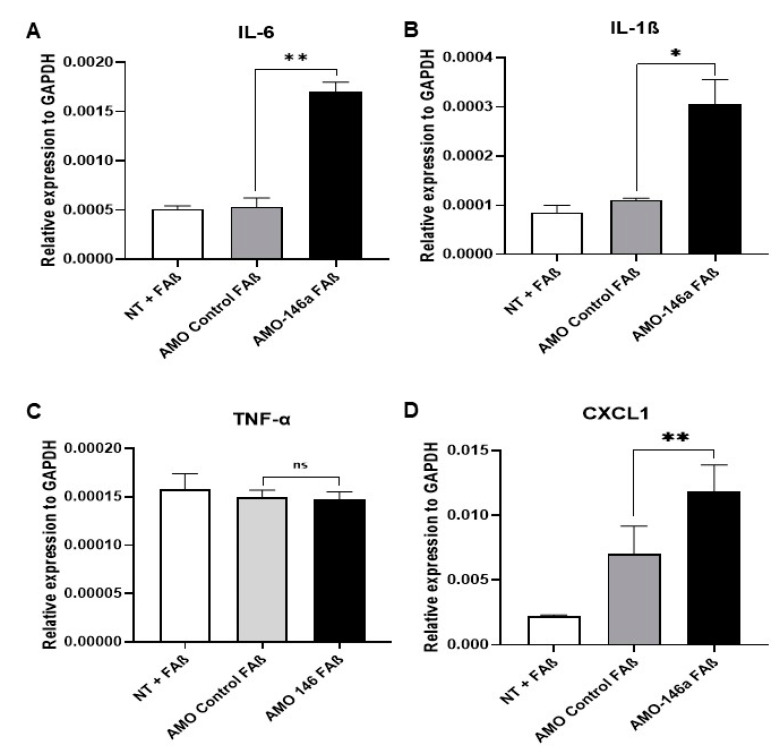
Transfection of miRNA-146-5p inhibitor in FAβ_1–42_-treated astrocytes rescues as well promote the expression of pro-inflammatory molecules. Astrocytes were first transfected with AMO-146a or AMO-CTL (100 nM), then treated with FAβ_1–42_ peptides (2 µM) before quantification relative of the expression of (**A**) IL-6, (**B**) IL-1β, (**C**) TNF-α, and (**D**) CXCL1 by qRT-PCR. Data are represented as mean ± SEM of one representative experiment performed twice in triplicate. Statistical comparisons between groups were made using a Student’s *t*-test. ** p* ≤ 0.05, ** *p* ≤ 0.01.

## Data Availability

Not applicable.

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
