# Peer review of "Intrahippocampal Inoculation of Aβ1–42 Peptide in Rat as a Model of Alzheimer’s Disease Identified MicroRNA-146a-5p as Blood Marker with Anti-Inflammatory Function in Astrocyte Cells"

_cells, 2023, doi:10.3390/cells12050694_

Round 1

Reviewer 1 Report

Please, find comments and suggestions for authors in the attached file

Author Response

REVIEWER 1

Comments and Suggestions for Authors

The manuscript presented by R Aquino et al. studies an interesting aspect in the field of Alzheimer’s disease, which is the need of finding good early biomarkers for the disease. The work is well planned and executed, although some points need to be addressed before the publication of the work.

  1. I would consider to change a little bit the title as, as it is written now, it seems that Ab is inoculated in a transgenic AD rat model or in a rat model that is already an AD I would suggest to replace, for example, “inthrahippocampal inoculation of Ab1-42 peptide in rat model of Alzheimer’s disease…” by “intrahippocampal inoculation of Ab1-42 in rats as a model of Alzheimer’s disease…”, to indicate that the model itself consists in the inoculation of Ab in WT animals. Otherwise, it is confusing.

We agree with the Reviewer 1 that the original title version could be confusing. We have changed the title as requested. This change can be read on page 1, line 1.

  1. In section 2, authors cite reference 30 when explaining the procedure for preparing fibrillar Abeta for in vivo inoculation. However, the method is not exactly the same (Ab40 vs Ab42, saline vs PBS, 1 week vs 5 days incubation…), and neither this cite, nor authors in this study provide evidences that the procedure gives indeed Abeta fibrils. In particular, Ab40 and Ab42 show very different aggregation properties. Therefore, authors should cite a reference that uses their same method and shows fibrils formation, or demonstrate by themselves the formation of fibrils using this method.

In our study, we prepared FAβ1-42 using a general procedure frequently used by different groups from Aβ1-42 peptides purchased from Sigma for example. The incubation of Aβ1-42 peptides for 5 days at 37°C is considered as an appropriate procedure for the assembly of Aβ peptides into fibrils while shorter the incubation period as 2 days and incubation peptide at 4°C is also a widely accepted procedure to generate oligomer form of Ab (OAβ) (Dahlgren et al., 2002). We infused the fibrillary form in our animals as this form induces significant change in hippocampal tissues as astrogliosis, change in miRNA expression and cognitive impairment. We reported the same molecular and cognitive change in our animal model and assigned a specific molecular signature of circulating miRNAs with the presence of FAβ1-42 into hippocampal tissues. However, to avoid confusion we have now referenced specific publications that used the same method to generate the fibrillar form of FAβ1-42. This change can be found on page 4, line 156.

  1. Lines 153-155: authors explain that the in vivo inoculation procedure consists in a 6 min infusion followed by a 3 min period to allow the dispersion of solutions into de ventricles. However, two lines after this authors explain that coordinates correspond to the CA1 region of the hippocampus, which is inconsistent with the dispersion of solutions into the ventricles, and seem to wait for 5 min before removing the syringe instead of 3 min cited before, which is confusing. Please, clarify whether injection was performed into the ventricles or in the CA1, and how much time and when was waited to allow the diffusion of the injected

We thank the Reviewer 1 for pointing out this in our material and methods. We have now clarified the method used to infuse the FAβ1-42 solution in the hippocampal of rats. This change can be found on page 4, line 165-172.

  1. Does this model end with accumulation of FAβ1-42  in form of plaques or similar?

In our study, we did not verify this point as all of our animals were anesthetized at day 21 to collect samples (blood, tissues,…). However, based on literature review in the field as Calvo-Flores Guzmán et al., 2020 for instance, it is unlikely that FAβ1-42 peptides persist in hippocampus tissues to form amyloid plaques plaque as these peptides are rapidly cleared from tissues.

  1. When describing the method used for collecting CSF samples, authors refer to the publication by Blanco et al (ref 35). However, this publication describes the optimization of the method by Liu et al, and both are performed in mice instead of rat. CSF extraction is complicated in mice due to the small anatomy of their cisterna magna, and I am not sure about the translationally of the method to Authors should better reference the method, citing appropriate bibliography that uses this method in rats. In addition, good and clear CSF extraction relies in most of cases in a very few amount extracted (going from 1- 2 to 5 µl in mice) in order to ensure its purity without contamination; authors should specify how much amount of CSF was approximately extracted in their rats, and whether it was contamined or not, in such case should be discarded for further analyses. The obtention of non-contamined CSF is the most important parameter to consider a good CSF sample to be analyzed, as normally, extraction of large volumes can bring with it contamination with small pieces of brain tissue or blood, which even performing centrifugation or clarification procedures can include molecules belonging to plasma. In this regard, authors should justify why they performed centrifugation to eliminate debris or clarified CSF samples, as well as provide evidences that their CSF was not contamined with plasma, for example by determining its content of catalase or any plasmatic protein.

Methods used to collect CSF samples in rats and mice do not differ but it is easier to perform the surgical procedure in rats due to their anatomy. As requested, we removed bibliography references describing methods used to prepare CSF in mice and kept in the manuscript revised text references citing methods to prepare CSF in rats. We routinely collected about 40 to 80 μL of CSF samples in our animals, which is fairly the same volume as described in the literature (Pegg et al., 2010, for instance). Experienced and skilled technicians, who are practicing this procedure routinely from more than 5 years, have performed the extraction of CSF samples for this study. Nevertheless, please note that for some animals, it was difficult and sometimes impossible to collect enough CSF. For other animals (the majority), we took precaution to not contaminate samples with blood during the extraction procedure. After a visual inspection, only clear samples were conserved for the following experiments while other samples were discarded. We subsequently centrifuged all samples as routine procedure for microRNA extraction and analysis. In spite of, please note that contaminated blood-samples cannot be processed for miRNA quantification as blood contents interfere with chemical reagents during the miRNA extraction procedure and make impossible the collection of a good Ct value during the qPCR reaction. We have clarified the procedure used to extract CSF from rats. This revision can be found on page 5, line 207-214.

  1. In section 8.2, please specify culture concentration conditions for LPS treatments.

As requested, we have specified culture conditions for LPS treatment in page 6, line 282.

  1. Authors should replace the term “non-familial AD disorder” in the abstract (lines 20 and 31-32) by “sporadic AD” or “LOAD”, as they refer to in the introduction, as these are the terms generally used to refer to the disease when it does not depend on known genetic mutations.

As requested, we have modified the text accordingly in page 1, line 37.

  1. Sentence in lines 396-397: add that these results were found in the second cohort, as the sentence is not very clear otherwise.

We thank the reviewer for highlighting this point. The change can be found on page 9, line 409.

  1. Why miR-125b-5p and miR-191-5p were not analyzed in the kinetics of rats treated at 2.5 ug/ul (results, section 3.5)?

Despite the fact that miRNA-125b-5p and miRNA-191-5p were detected as the most deregulated miRNAs in our animal model of ADwe decided not to include these two miRNAs in the kinetic study for several reasons. First, these two miRNAs were only detected in the second cohort of animal who received the highest dose of FAβ peptides solution. Therefore, these two miRNAs are less prone to be good candidates for (very) early diagnostic markers capable to discriminate individuals containing small traces of Aβ peptide in the brain. Secondly, because these two miRNAs are widely described in literature and more closely related to Tau hyper phosphorylation, event that is subsequent to Aβ peptide aggregation that we did not study in this work. Third, we focused on miRNA-9, -29a, -29c and -146a-5p as this set of miRNAs are less described in the literature whilst they were detected in the two cohorts of animals receiving the low and high dose of FAβ peptide. Nevertheless, we believe, as the Reviewer 1, that these 2 miRNAs, miRNA-125b-5p & miRNA-191-5p, deserve further investigation in the future, particularly in this FAβ-infused animal model of AD.

  1. Related to the kinetics of circulating miRNAs in rats infused with FAb 1- 42 at 2.5 ug/ul, graph from miR-146a-5p (Fig 5A), and to a lesser extent, miR- 29a-3p and miR-29c-3p (Fig 5 B,C) show increasing values in the control group through time, while one would expect them to remain more stable, as no treatment is infused to these animals (other than PBS). Particularly in case of miR-146a-5p, values from Fabeta-infused rats remain very close to basal values, except for day 14. Do authors have any explanation for this effect? Could the surgery be involved, maybe explaining why this miRNA is not altered in the APP/PS1 mouse model, not being subjected to surgery? This needs to be

We grateful the Reviewer 1 for this constructive comment that highlights an important point. We completely agree that in control group of rats there is a trend for a general increase in expression of the three miRNAs after infusion the FAβ peptides solution. This might reflect, as anticipated by the Reviewer 1, an inflammation reaction generated by the surgical procedure to deliver the FAβ peptides into the hippocampal of animal. This observation is not entirely surprising as miRNAs regulate many, if not all of biological processes, including inflammatory reactions generated by a perfusion system. Therefore, it cannot be excluded that at early time point after the surgery, the expression of miRNA-146-5p induced by presence of FAβ peptides could be masked by the inflammatory reactions induced by the surgery and for which the miRNA-146-5p is also expected to be regulated. This might indeed explained the absence of differential expression of miRNA-146-5p at day 7 between the control and FAβ-infused animals. However, and beyond this “noise of expression” induced by the surgery, we found statistically significant differences in terms of miRNA-146a-5p expression at later time point between the two groups of animals (PBS infused rats versus FAβ peptides infused rats). This means that the statistically significant differences reported in the Figure 5A at day 14 and 21 account for the presence of FAβ1-42 peptides rather than a general reaction induced by the surgery. This animal model of intrahippocampal infusion of Aβ1-42 is widely used (Facchinetti et al., 2018; Wong et al., 2016; Karthick et al., 2019) and is considered relevant when appropriate controls are implemented as we did in our study.

Concerning the discrepancy of miRNA-146a-5p expression between the two animal models used in our study, please note that the Reviewer 2 has also raised this point. As this point is of importance for the global understanding of our work, we have added a comment in the discussion part of the revised version of our manuscript. The comment can be found in page 19, line 693 and can be read as fellow:

This discrepancy between these 2 animal models of AD can be explained by the fact that the FAβ1-42 infusion model represents a model of inflammation induced by the direct ex-position of hippocampal tissues to FAβ1-42 peptides load [71] while the APPswe/PS1dE9 transgenic model corresponds to a model of chronic inflammation induced by the gradual accumulation of Aβ deposits over the lifespan of transgenic animals, which express constitutive mutant forms of APP and PS1 [72]. Therefore, the inflammatory responses are expected to be different in these 2 models and should implicate different cellular pathways and molecular partners. This difference might be also accentuated by compensatory or adaptive responses developed in knocking or knockout transgenic animals [73,74].

71 : McLarnon, J.G.; Ryu, J.K. Relevance of abeta1-42 intrahippocampal injection as an animal model of inflamed Alzheimer's disease brain. Current Alzheimer research 2008, 5, 475-480, doi:10.2174/156720508785908874.

72 : Huang, H.; Nie, S.; Cao, M.; Marshall, C.; Gao, J.; Xiao, N.; Hu, G.; Xiao, M. Characterization of AD-like phenotype in aged APPSwe/PS1dE9 mice. Age (Dordrecht, Netherlands) 2016, 38, 303-322, doi:10.1007/s11357-016-9929-7..

73 : El-Brolosy, M.A.; Stainier, D.Y.R. Genetic compensation: A phenomenon in search of mechanisms. PLoS genetics 2017, 13, e1006780, doi:10.1371/journal.pgen.1006780.

74 : Sztal, T.E.; Stainier, D.Y.R. Transcriptional adaptation: a mechanism underlying genetic robustness. Development (Cambridge, England) 2020, 147, doi:10.1242/dev.186452.

  1. In fig 9, IL-6, IL-1b and TNFa should be quantified by ELISA instead of mRNA to confirm the absence of changes, as this would be much more informative about the status of cytokines release than mRNA

We agree with Reviewer 1 that ELISA quantification of cytokines IL-6, IL-1b and TNFa would have been informative as well. However, we used LPS as positive control and found a statistically significant induction of IL-6 and IL-1b expression as found by others (Cui et al., 2010; Li et al., 2011; Lukiw et al., 2008). Furthermore, and as mentioned in the Discussion part of our manuscript, it was reported that treatment of neuronal or astrocyte with Aβ peptides alone, without addition of pro-inflammatory molecules, is not sufficient to induce the production of cytokines as we found here in our study. Therefore, we thought that dosing cytokines production by the ELISA method would not add an important plus value to this work.

  1. Figure 10B should contain values from non-treated cells as a reference for total (or partial) recovery after treatment with AMO-146.

Thank you for raising these missing points. As requested, we have now included in our Figure graph the experimental values generated previously with non-treated astrocyte (NT). Results shown in the modified Figure 10 below indicate that AMO-146a rescues not only expression of IRAK-1 in these cells but also induces a significant overexpression of IRAK-1 and TRAF-6, superior to the steady stale level of expression detected in non-treated astrocytes. This latest result tends to reinforce our hypothesis that miR-146 exerts a negative feedback loop over the NFkb pathway and that interfering its expression, and therefore this biological function, might push the cells to a pro-inflammatory state by deregulating expression of IRAK-1 and TRAF-6. It is expected that the overexpression of TRAF-6 allows it to reach a threshold level that activates upstream molecular mediators of the NF-κb and ultimately production of inflammatory cytokines. As this point is also of importance for the global understanding of our work, we have added a comment in the discussion part of the revised version of our manuscript. The comment can be found on page 20, line 764.

Figure 10. Transfection of miR-146a inhibitor in FAβ1-42-treated astrocyte cells rescues expression of IRAK-1 and TRAF-6 but not IRAK-2. A) qRT-PCR evaluation of performance of anti-miR-146a oligonucleotides (AMO-146a) to down regulate expression of miR-146a in astrocyte cells. B) qRT-PCR evaluation of impact of AMO-146 transfection on expression of transcriptional targets of miR-146a in FAβ1-42-astrocyte treated cells. Cells were first transfected with AMO-146a or AMO-CTL (100 nM) then treated with FAβ1-42 peptides solutions (2 µM) before quantification of the relative expression of IRAK1/2 and TRAF-6 transcriptional targets of miR-146a. Non-treated-astrocytes (NT, no FAβ1-42 treatment) were used as additional control. Data are represented as mean ± SEM of one representative experiment performed twice in triplicate. Statistical comparisons between groups were made using the Student's t-test. * p ≤ 0.05, ** p ≤ 0.01, *** p≤ 0.001.

  1. Line 640, Fig 4C and 4D should be Fig 5C and

We thank the Reviewer 1 for noticing this typo and error. We have corrected the text accordingly. The change can be read on page 18, line 661.

  1. Sentence in lines 719-722 is difficult to follow and to Please, rewrite it to be more clear.

As requested, we clarified our statement and rewrite the sentence.  The change can be read on page 20, lines 773 to 779.

  1. NF-Kb activation in inflammatory cells or in neurons may activate different pathways, with different consequences, and these cell types are in the brain subjected to the same Abeta Why authors chosed to use astrocytes and not microglia for the analysis of the inflammatory capacities of miR-146a-5p? Could authors speculate about the effect that their findings would have in neurons?

We chose to perform our mechanistic study with astrocyte for several reasons. The first is that we sought to recapitulate as closer as possible the AD-like environment induced by infusion of FAβ peptides in hippocampal tissues in which astrocytes are one of most abundant cell types. Secondly, because astrocytes are considered to promote the first line of inflammatory responses in the CNS by regulating the expression of mediators of acute, innate and adaptive immune responses. Therefore, we think that investigating the anti-inflammatory function of miRNA-146a-5p is astrocytes makes sense and is appropriate. We anticipate similar cellular responses in neuronal cell as this cell type also prime inflammatory and immune responses (Li et al., 2011). Of note, some reports have shown that biological function of miRNA 146 in neurons is related to regulation of the tetraspanin 12 (TSPAN12), which serves as an essential partner for ADAM10, facilitating ADAM10-dependent proteolysis of the beta- amyloid precursor protein APP/PS1 (APP) in non-amyloidogenic signaling pathways.

  1. Please, ammend multiple spelling/gramar errors, some of them here showed but a complete english revision should be performed:

We thank the Reviewer 1 for noticing these typo and errors that we have corrected in the revised version of the manuscript.

  1. Line 70: “easy to detected” should be “easy to detect” or “easy to be detected”, and “…expression pattern that reflect…” should be “…reflects…”
  2. Line 138: “used” should be “use”
  3. Line 180: something is missing/misplaced in the sentence “The objective of this trial if to record…”
  4. Line 263: “approximatively” should be “approximately”
  5. Line 287: “expect” should be “except”
  6. Line 443: “animal” should be “animals”
  7. Line 653: the “to” a the end of the sentence does not make sense.
  8. Line 676-677: “inflammatory” should be “inflammation” or “and” should be deleted

BIBLIOGRAPHY REFERENCES

  1. Dahlgren, K.N.; Manelli, A.M.; Stine, W.B., Jr.; Baker, L.K.; Krafft, G.A.; LaDu, M.J. Oligomeric and fibrillar species of amyloid-beta peptides differentially affect neuronal viability. The Journal of biological chemistry 2002, 277, 32046-32053, doi:10.1074/jbc.M201750200.

  1. Calvo-Flores Guzmán, B.; Elizabeth Chaffey, T.; Hansika Palpagama, T.; Waters, S.; Boix, J.; Tate, W.P.; Peppercorn, K.; Dragunow, M.; Waldvogel, H.J.; Faull, R.L.M.; et al. The Interplay Between Beta-Amyloid 1-42 (Aβ(1-42))-Induced Hippocampal Inflammatory Response, p-tau, Vascular Pathology, and Their Synergistic Contributions to Neuronal Death and Behavioral Deficits. Frontiers in molecular neuroscience 2020, 13, 522073, doi:10.3389/fnmol.2020.552073.

  1. Pegg, C.C.; He, C.; Stroink, A.R.; Kattner, K.A.; Wang, C.X. Technique for collection of cerebrospinal fluid from the cisterna magna in rat. Journal of neuroscience methods 2010, 187, 8-12, doi:10.1016/j.jneumeth.2009.12.002.

  1. Facchinetti, R.; Bronzuoli, M.R.; Scuderi, C. An Animal Model of Alzheimer Disease Based on the Intrahippocampal Injection of Amyloid β-Peptide (1-42). Methods in molecular biology (Clifton, N.J.) 2018, 1727, 343-352, doi:10.1007/978-1-4939-7571-6_25.

  1. Wong, R.S.; Cechetto, D.F.; Whitehead, S.N. Assessing the Effects of Acute Amyloid β Oligomer Exposure in the Rat. International journal of molecular sciences 2016, 17, doi:10.3390/ijms17091390.

  1. Karthick, C.; Nithiyanandan, S.; Essa, M.M.; Guillemin, G.J.; Jayachandran, S.K.; Anusuyadevi, M. Time-dependent effect of oligomeric amyloid-β (1-42)-induced hippocampal neurodegeneration in rat model of Alzheimer's disease. Neurological research 2019, 41, 139-150, doi:10.1080/01616412.2018.1544745.

  1. Sztal, T.E.; Stainier, D.Y.R. Transcriptional adaptation: a mechanism underlying genetic robustness. Development (Cambridge, England) 2020, 147, doi:10.1242/dev.186452.

  1. El-Brolosy, M.A.; Stainier, D.Y.R. Genetic compensation: A phenomenon in search of mechanisms. PLoS genetics 2017, 13, e1006780, doi:10.1371/journal.pgen.1006780.

  1. Cui, J.G.; Li, Y.Y.; Zhao, Y.; Bhattacharjee, S.; Lukiw, W.J. Differential regulation of interleukin-1 receptor-associated kinase-1 (IRAK-1) and IRAK-2 by microRNA-146a and NF-kappaB in stressed human astroglial cells and in Alzheimer disease. The Journal of biological chemistry 2010, 285, 38951-38960, doi:10.1074/jbc.M110.178848.

  1. Li, Y.Y.; Cui, J.G.; Dua, P.; Pogue, A.I.; Bhattacharjee, S.; Lukiw, W.J. Differential expression of miRNA-146a-regulated inflammatory genes in human primary neural, astroglial and microglial cells. Neuroscience letters 2011, 499, 109-113, doi:10.1016/j.neulet.2011.05.044.

  1. Lukiw, W.J.; Zhao, Y.; Cui, J.G. An NF-kappaB-sensitive microRNA-146a-mediated inflammatory circuit in Alzheimer disease and in stressed human brain cells. The Journal of biological chemistry 2008, 283, 31315-31322, doi:10.1074/jbc.M805371200.

  1. De, S.; Wirthensohn, D.C.; Flagmeier, P.; Hughes, C.; Aprile, F.A.; Ruggeri, F.S.; Whiten, D.R.; Emin, D.; Xia, Z.; Varela, J.A.; et al. Different soluble aggregates of Aβ42 can give rise to cellular toxicity through different mechanisms. Nature communications 2019, 10, 1541, doi:10.1038/s41467-019-09477-3.

  1. Xu, Y.R.; Lei, C.Q. TAK1-TABs Complex: A Central Signalosome in Inflammatory Responses. Frontiers in immunology 2020, 11, 608976, doi:10.3389/fimmu.2020.608976.

  1. Saba, R.; Sorensen, D.L.; Booth, S.A. MicroRNA-146a: A Dominant, Negative Regulator of the Innate Immune Response. Frontiers in immunology 2014, 5, 578, doi:10.3389/fimmu.2014.00578.

Reviewer 2 Report

Aquino et al explored the role of microRNA in their article “Intrahippocampal inoculation of Aβ1-42 peptide in rat model of Alzheimer disease identified microRNA-146a as blood marker with anti-inflammatory function in astrocyte cells”. This work is interesting and their attempt to identify novel microRNA-based markers in this disease would be beneficial to develop AD diagnosis in its early stage of onset. The paper is very elaborate. The method section is descriptive with all details. Although the work presented here is of general interest, there is a good scope for improvement. Authors need to work on writing. There are several places where either sentence is incomplete or not making sense. Please see specific comments below. Hope these comments will be useful to improve the manuscript further.

1.     Last paragraph of the introduction mentioned testing various pre-reported miRNAs (line 109-15). It will be good to have a line mentioning about the identification of miR146 before discussing the possible role/association of this miRNA in cellular events.

2.     line 141: Please check the stock solution preparation of peptide. Initially, it is mentioned that 2 mM stock is prepared in DMSO. Then, I assume that DMSO soluble peptide was used to make stock solution in PBS but how can a 2 mM solution make 2.22 mM stock? Please rewrite the whole paragraph to make it clear.

3.     Line 266: I think the authors used 0.5 uM conc of FAB-1-42 peptide in the study. Please double-check the conc. it should be 0.5, not 5 uM. 

4.     it's been reported that FAB1-42 causes Tau phosphorylation and converts them into a toxic form. Did authors monitor the effect of the peptide by monitoring tau phosphorylation level, aggregation, and neuron viability upon peptide inoculation in rats? It is recommended to have WB image or immuno-histochemistry data included in the study to first established the peptide effect.

5.     Authors have used three different conc. of peptide in this study, and they found that 2.5 ug/ul is best in showing the considerable change in microRNA level in brain tissue. it will be good to have an image showing the level of GFAP level in astrocytes at this concentration. It could be possible that at this concentration there are more significant changes in GFAP level.

6.     section 3.4 title; peptides treatment is not enhancing but downregulating the level of various micro-RNAs. make correction.

7.     Line 493: check sentence. need to improve the writing.

8.     Line 505: What conc of the peptide was used here?

9. It is well known that APPswe/PS1dE9 transgenic mouse model also produces AB1-42 deposits, in that case, what could be the reason behind the unchanged expression of miR-146a expression in this mouse model compared to AB1-42 infusion? 

10.  Although authors have shown differential expression of mir-146a in CSF and serum, still it does not explain the differential expression of miR-146a-5p effect in the serum of two AD models. 

11.  IRAK1/IRAK2 both are downstream to NF-kB then what could be the reason behind the unchanged expression of IRAK2?

12.  Section 3.9; In AB1-42 peptide-treated astrocyte cells, TRAF6 remains unchanged then how did the author hypothesize that interfering expression of miR-146a-5p0 might rescue TRAF6 expression?

13.  In AB1-42 treated cells, there is no change in the expression of TRAF6 but AMO-146a rescued TRAF6 to the base level, what could be the possible explanation? These experiments and their interpretation need some attention?

Author Response

REVIEWER 2

Comments and Suggestions for Authors

Aquino et al explored the role of microRNA in their article “Intrahippocampal inoculation of Aβ1-42 peptide in rat model of Alzheimer disease identified microRNA-146a as blood marker with anti-inflammatory function in astrocyte cells”. This work is interesting and their attempt to identify novel microRNA-based markers in this disease would be beneficial to develop AD diagnosis in its early stage of onset. The paper is very elaborate. The method section is descriptive with all details. Although the work presented here is of general interest, there is a good scope for improvement. Authors need to work on writing. There are several places where either sentence is incomplete or not making sense. Please see specific comments below. Hope these comments will be useful to improve the manuscript further.

  1. 1.     Last paragraph of the introduction mentioned testing various pre-reported miRNAs (line 109-15). It will be good to have a line mentioning about the identification of miR146 before discussing the possible role/association of this miRNA in cellular events.

We thank the Reviewer 2 and agree with that mentioning the rational choice of miRNA-146a-5p will make the introductory section easier to read and will benefit the overall understanding of our work. We have modified the text accordingly. The change can be read on page 3, line 120-123

  1. line 141: Please check the stock solution preparation of peptide. Initially, it is mentioned that 2 mM stock is prepared in DMSO. Then, I assume that DMSO soluble peptide was used to make stock solution in PBS but how can a 2 mM solution make 2.22 mM stock? Please rewrite the whole paragraph to make it clear.

We apologize and thank the Reviewer 2 for noticing this typo and error. We have corrected the text accordingly. The change can be read on page 4, line 152.

  1. Line 266: I think the authors used 0.5 uM conc of FAB-1-42 peptide in the study. Please double-check the conc. it should be 0.5, not 5 uM. 

We apologize and thank the Reviewer 2 for noticing again this typo and error. We have corrected the text accordingly. The change can be read on page 6, line 278.

  1. It has been reported that FAB1-42 causes Tau phosphorylation and converts them into a toxic form. Did authors monitor the effect of the peptide by monitoring tau phosphorylation level, aggregation, and neuron viability upon peptide inoculation in rats? It is recommended to have WB image or immuno-histochemistry data included in the study to first established the peptide effect.

We agree with the Reviewer 2 that monitoring Tau phosphorylation level, aggregation and neuron viability upon peptide inoculation in rats will add a plus value on our manuscript. However, studying the impact of FAβ1-42 deposition on Tau phosphorylation in our animal model of AD will require an additional large cohort of animals as well as significant experimental procedures as brain biopsies for immunohistochemically analysis for instance. All of these procedures, beyond being time- and money-consuming that we can unfortunately not afford, could not objectively fall within the principal scope of our manuscript.

  1. Authors have used three different conc. of peptide in this study, and they found that 2.5 ug/ul is best in showing the considerable change in microRNA level in brain tissue. it will be good to have an image showing the level of GFAP level in astrocytes at this concentration. It could be possible that at this concentration there are more significant changes in GFAP level.

We also agree that it will be great to have on image showing the change GFAP expression upon infusion of two-times more FAβ1-42 peptides in brain of our animals. However, addressing this point will require significant additional in vivo and in situ experiments that will not change the main message of this manuscript part that is : FAβ1-42 infusion leads to an inflammatory response in the hippocampus of rats as attested by GFAP staining.

  1. 6.     section 3.4 title; peptides treatment is not enhancing but downregulating the level of various micro-RNAs. make correction.

We apologize for this typo error that we corrected on page 9, line 399.

  1. Line 493: check sentence. need to improve the writing.

We have improved the writing as requested. The change can be read on page 13, line 507.

  1. Line 505: What conc of the peptide was used here?

We have now specified the concentration of peptide solution used. The change can be found on page 13, line 517.

  1. It is well known that APPswe/PS1dE9 transgenic mouse model also produces AB1-42deposits, in that case, what could be the reason behind the unchanged expression of miR-146a expression in this mouse model compared to AB1-42 infusion?

We agree that it might be surprising to note that miRNA-146a-5p was not detected as differentially dysregulated in the APPswe/PS1dE9 transgenic mice model of AD although these mice have Aβ deposits according to the literature. Reviewer 1 raised also this comment. As this point is of importance for the global understanding of our work, we have added a comment in the discussion part of the revised version of our manuscript. The comment can be found in page 19, line 693 and can be read as fellow:

This discrepancy between these 2 animal models of AD can be explained by the fact that the FAβ1-42 infusion model represents a model of inflammation induced by the direct ex-position of hippocampal tissues to FAβ1-42 peptides load [71] while the APPswe/PS1dE9 transgenic model corresponds to a model of chronic inflammation induced by the gradual accumulation of Aβ deposits over the lifespan of transgenic animals, which express constitutive mutant forms of APP and PS1 [72]. Therefore, the inflammatory responses are expected to be different in these 2 models and should implicate different cellular pathways and molecular partners. This difference might be also accentuated by compensatory or adaptive responses developed in knocking or knockout transgenic animals [73,74].

71 : McLarnon, J.G.; Ryu, J.K. Relevance of abeta1-42 intrahippocampal injection as an animal model of inflamed Alzheimer's disease brain. Current Alzheimer research 2008, 5, 475-480, doi:10.2174/156720508785908874.

72 : Huang, H.; Nie, S.; Cao, M.; Marshall, C.; Gao, J.; Xiao, N.; Hu, G.; Xiao, M. Characterization of AD-like phenotype in aged APPSwe/PS1dE9 mice. Age (Dordrecht, Netherlands) 2016, 38, 303-322, doi:10.1007/s11357-016-9929-7..

73 : El-Brolosy, M.A.; Stainier, D.Y.R. Genetic compensation: A phenomenon in search of mechanisms. PLoS genetics 2017, 13, e1006780, doi:10.1371/journal.pgen.1006780.

74 : Sztal, T.E.; Stainier, D.Y.R. Transcriptional adaptation: a mechanism underlying genetic robustness. Development (Cambridge, England) 2020, 147, doi:10.1242/dev.186452.

  1. Although authors have shown differential expression of mir-146a in CSF and serum, still it does not explain the differential expression of miR-146a-5p effect in the serum of two AD models. 

We have addressed this point in the comment above.

  1. IRAK1/IRAK2 both are downstream to NF-kB then what could be the reason behind the unchanged expression of IRAK2?

As stated by the Reviewer 2, NF-κB cell signalling pathway is dependent to downstream intracellular mediators such as IRAK1/2 and TRAF-6. Upon stimulation of Toll-like receptors (TLRs) and particularly TLR4, which has been recently assigned as principal receptor of Aβ peptides (De et al., 2019), there is recruitment of the adaptor protein MYD88 which recruits IRAK-1 and IRAK-2 to form a multifunctional supramolecular organizing center termed myddosome. This myddosome complex activates subsequently many molecular cascades of events as TRAF-6 dimerization and ubiquitination, activation of protein kinases TAK1 and TAB1 and IκB-α phosphorylation for nuclear translocation of the p55/65 submits and finally transcription of inflammatory cytokines (Xu and Lei, 2021). Of note, IRAK-1, -2 and TRAF-6 mRNAs harbour binding sequence to miRNA-146, yielding the hypothesis that miRNA-146a-5p might regulate NF-κB cell-signalling pathway (Saba et al., 2014). While the biological role of IRAK-1 on this signalling is consensual on the literature, the signalling role of IRAK-2 is not and is a matter of debate. Some authors (Saba et al., 2014) have reported a compensatory mechanism between these two targets, meaning that downregulation of IRAK-1 by miRNA-146a-5p in cells leads to upregulation of IRAK-2. Other studies reported that miRNA-146a-5p modulates the expression of IRAK-1 but not IRAK-2, suggesting the existence of an independent regulatory mechanism between these two targets to control the steady state level of the IRAK-1 and -2 (Cui et al., 2010). Based on this finding and data in the Figure 8 showing that treatment of astrocyte with FAβ peptides do not change expression of IRAK-2, it can be concluded that expression of IRAK-2 is maintained in astrocytes through an independent regulatory mechanism that remains to be identified and beyond the scope of the present study.

  1. Section 3.9; In AB1-42peptide-treated astrocyte cells, TRAF6 remains unchanged then how did the author hypothesize that interfering expression of miR-146a-5p might rescue TRAF6 expression?

The observation that the downregulation IRAK-1 did not change the expression of TRAF-6 and did not induce as well production of cytokines in FAβ-treated astrocytes let us to hypothesize that interfering with miRNA-146a-5p expression would unbalance the steady state level of TRAF-6 expression and, as consequence should promote the overall activation of the upstream NFkB signalling cascade of events described above in point 11. Our results shown in the Figure 10B tend to support this hypothesis. Transfection of astrocytes with the AMO-146 before incubation with FAβ peptides induced a statistically significant overexpression of IRAK-1 and TRAF-6, superior to the “basal” expression level detected in astrocytes treated with FAb alone. It can then be anticipated that this elevated level of TRAF-6 allows it to reach a threshold that activates upstream molecular mediators of the NF-κB and ultimately production of inflammatory cytokines. The observation that pro-inflammatory cytokines IL-6, IL-1β and CXCL1 are induced in AMO-146 treated astrocytes support this hypothesis.

  1. In AB1-42 treated cells, there is no change in the expression of TRAF6 but AMO-146a rescued TRAF6 to the base level, what could be the possible explanation? These experiments and their interpretation need some attention?

We have not explored the exact mechanisms by which TRAF-6 overexpression led to the production of cytokines in astrocytes treated with AMO-146 & FAβ peptides. As mentioned above, there are many mediators of the NFkB pathway that might be activated in response to inhibition of miRNA-146a. We do not exclude a well as participation of other cell signalling pathways. Investigating in deep the interplay between miRNA-146a-5p, IRAK-1/2, TRAF-6 expression in the production of cytokines requires a complete program of work and therefore could hardly fall within the scope of our present manuscript. Please note that the principal aim of our study was to report a comprehensive work-up procedure to 1) identify a panel of circulating blood miRNAs associated with the presence of FAβ1-42 peptides into hippocampus of rats for diagnosis purposes; 2) to perform a comparative study with the APPswe/PS1dE9 transgenic mouse model of Alzheimer's disease to more precisely define the mode of induction of miRNA-146a-5p in response to FAb infusion and 3) to delineate the biological function miRNA-146a-5p in astrocyte treated with FAβ1-42 peptides.

BIBLIOGRAPHY REFERENCES

  1. Dahlgren, K.N.; Manelli, A.M.; Stine, W.B., Jr.; Baker, L.K.; Krafft, G.A.; LaDu, M.J. Oligomeric and fibrillar species of amyloid-beta peptides differentially affect neuronal viability. The Journal of biological chemistry 2002, 277, 32046-32053, doi:10.1074/jbc.M201750200.

  1. Calvo-Flores Guzmán, B.; Elizabeth Chaffey, T.; Hansika Palpagama, T.; Waters, S.; Boix, J.; Tate, W.P.; Peppercorn, K.; Dragunow, M.; Waldvogel, H.J.; Faull, R.L.M.; et al. The Interplay Between Beta-Amyloid 1-42 (Aβ(1-42))-Induced Hippocampal Inflammatory Response, p-tau, Vascular Pathology, and Their Synergistic Contributions to Neuronal Death and Behavioral Deficits. Frontiers in molecular neuroscience 2020, 13, 522073, doi:10.3389/fnmol.2020.552073.

  1. Pegg, C.C.; He, C.; Stroink, A.R.; Kattner, K.A.; Wang, C.X. Technique for collection of cerebrospinal fluid from the cisterna magna in rat. Journal of neuroscience methods 2010, 187, 8-12, doi:10.1016/j.jneumeth.2009.12.002.

  1. Facchinetti, R.; Bronzuoli, M.R.; Scuderi, C. An Animal Model of Alzheimer Disease Based on the Intrahippocampal Injection of Amyloid β-Peptide (1-42). Methods in molecular biology (Clifton, N.J.) 2018, 1727, 343-352, doi:10.1007/978-1-4939-7571-6_25.

  1. Wong, R.S.; Cechetto, D.F.; Whitehead, S.N. Assessing the Effects of Acute Amyloid β Oligomer Exposure in the Rat. International journal of molecular sciences 2016, 17, doi:10.3390/ijms17091390.

  1. Karthick, C.; Nithiyanandan, S.; Essa, M.M.; Guillemin, G.J.; Jayachandran, S.K.; Anusuyadevi, M. Time-dependent effect of oligomeric amyloid-β (1-42)-induced hippocampal neurodegeneration in rat model of Alzheimer's disease. Neurological research 2019, 41, 139-150, doi:10.1080/01616412.2018.1544745.

  1. Sztal, T.E.; Stainier, D.Y.R. Transcriptional adaptation: a mechanism underlying genetic robustness. Development (Cambridge, England) 2020, 147, doi:10.1242/dev.186452.

  1. El-Brolosy, M.A.; Stainier, D.Y.R. Genetic compensation: A phenomenon in search of mechanisms. PLoS genetics 2017, 13, e1006780, doi:10.1371/journal.pgen.1006780.

  1. Cui, J.G.; Li, Y.Y.; Zhao, Y.; Bhattacharjee, S.; Lukiw, W.J. Differential regulation of interleukin-1 receptor-associated kinase-1 (IRAK-1) and IRAK-2 by microRNA-146a and NF-kappaB in stressed human astroglial cells and in Alzheimer disease. The Journal of biological chemistry 2010, 285, 38951-38960, doi:10.1074/jbc.M110.178848.

  1. Li, Y.Y.; Cui, J.G.; Dua, P.; Pogue, A.I.; Bhattacharjee, S.; Lukiw, W.J. Differential expression of miRNA-146a-regulated inflammatory genes in human primary neural, astroglial and microglial cells. Neuroscience letters 2011, 499, 109-113, doi:10.1016/j.neulet.2011.05.044.

  1. Lukiw, W.J.; Zhao, Y.; Cui, J.G. An NF-kappaB-sensitive microRNA-146a-mediated inflammatory circuit in Alzheimer disease and in stressed human brain cells. The Journal of biological chemistry 2008, 283, 31315-31322, doi:10.1074/jbc.M805371200.

  1. De, S.; Wirthensohn, D.C.; Flagmeier, P.; Hughes, C.; Aprile, F.A.; Ruggeri, F.S.; Whiten, D.R.; Emin, D.; Xia, Z.; Varela, J.A.; et al. Different soluble aggregates of Aβ42 can give rise to cellular toxicity through different mechanisms. Nature communications 2019, 10, 1541, doi:10.1038/s41467-019-09477-3.

  1. Xu, Y.R.; Lei, C.Q. TAK1-TABs Complex: A Central Signalosome in Inflammatory Responses. Frontiers in immunology 2020, 11, 608976, doi:10.3389/fimmu.2020.608976.

  1. Saba, R.; Sorensen, D.L.; Booth, S.A. MicroRNA-146a: A Dominant, Negative Regulator of the Innate Immune Response. Frontiers in immunology 2014, 5, 578, doi:10.3389/fimmu.2014.00578.

Round 2

Reviewer 2 Report

The new version of the manuscript looks better and improved. All the minor comments are addressed to satisfaction.  I will suggest including the explanations answered for “point number 13” in my original comment in the discussion or conclusion section.

Author Response

To the Editorial Board, Cells Journal

Special Issue "microRNA as Biomarker II"

Dear Professor Taguchi,

Dear Katya Su, Section Managing Editor, MDPI

You will find enclosed our revised version 2 of our manuscript entitled “Intrahippocampal inoculation of Aβ1-42 peptides as a rat model of Alzheimer disease, identified microRNA-146a as blood marker with anti-inflammatory function in astrocyte cells ", by Aquino et al., for publication in the special Issue “microRNA as Biomarker II”.

As requested by the Reviewer 2, we have included a comment in the discussion part of the revised version of our manuscript in page 20. On page 21, we also acknowledge our colleague, Dr. Denoyelle for his help in setting up the sensitive method used in our study to detect circulating miRNAs in blood samples. The change are visible on track change format and are highlighted in yellow in this second revision text.

We are happy that Editors and Reviewers appreciate change made on our manuscript. We hope that our paper will now be considered for publication in Cells.

 Yours sincerely,

Patrick Baril, Ph.D

Co-corresponding author